# Translation of small downstream ORFs enhances translation of canonical main open reading frames

Qiushuang Wu[1],[†] (ID), Matthew Wright[1],[†] (ID), Madelaine M Gogol[1], William D Bradford[1], Ning Zhang[1] & Ariel A Bazzini[1,2],[*] (ID)

## Abstract

In addition to canonical open reading frames (ORFs), thousands of translated small ORFs (containing less than 100 codons) have been identified in untranslated mRNA regions (UTRs) across eukaryotes. Small ORFs in 5′ UTRs (upstream (u)ORFs) often repress translation of the canonical ORF within the same mRNA. However, the function of translated small ORFs in the 3′ UTRs (downstream (d)ORFs) is unknown. Contrary to uORFs, we find that translation of dORFs enhances translation of their corresponding canonical ORFs. This translation stimulatory effect of dORFs depends on the number of dORFs, but not the length or peptide they encode. We propose that dORFs represent a new, strong, and universal translation regulatory mechanism in vertebrates.

**Keywords** dORF; ribosome profiling; translation efficiency

**Subject Categories** Chromatin, Transcription & Genomics; Translation & Protein Quality

**The EMBO Journal (2020) 39: e104763**

See also: **S Dodbele & JE Wilusz** (September 2020)

## Introduction

The dogma that each eukaryotic messenger RNA (mRNA) encodes a single protein has undergone a revision in recent years. Ribosome and proteomic profiling have revealed many small, translated open reading frames (ORFs) within regions of mRNAs that were previously designated "untranslated regions" (UTRs) as well as in long non-coding RNAs from multiple species and viruses (Slavoff *et al*, 2013; Bazzini *et al*, 2014; Stern-Ginossar & Ingolia, 2015; Calviello *et al*, 2016; Couso & Patraquim, 2017; Makarewich & Olson, 2017; Brunet *et al*, 2018). In some cases, the peptides produced by such ORFs have been shown to play functional

roles. For example, myoregulin, a conserved 46-amino acid peptide in human and mice encoded by a skeletal muscle-specific RNA previously annotated as a long non-coding RNA, regulates muscle performance (Anderson *et al*, 2015). In many cases, however, translation of the ORF *per se* has a regulatory function that is independent of the peptide produced (Barbosa *et al*, 2013; Couso & Patraquim, 2017). For example, translation of small open reading frames in the 5′ UTR—called upstream (u)ORFs—often decreases translation efficiency of the canonical ORF (Mueller & Hinnebusch, 1986; Vattem & Wek, 2004; Brar *et al*, 2012; von Arnim *et al*, 2014; Wethmar *et al*, 2014; Chew *et al*, 2016; Johnstone *et al*, 2016). Consistent with uORF translation serving important regulatory roles, uORFs are pervasive from yeast to human and their presence is conserved between orthologous genes despite having little similarity at the level of amino acid sequence (Chew *et al*, 2016; Johnstone *et al*, 2016; Dumesic *et al*, 2019). For example, fundamental developmental genes such as *POU5F3* (*Oct4*), *Nanog*, and *Smad7* encode multiple uORFs from zebrafish to human but the amino acid identity encoded by the uORF is not conserved (Johnstone *et al*, 2016). Both ATG and the alternative (non-ATG) translation start codons, CTG, GTG, and TTG, have been identified in uORFs (Brar *et al*, 2012; Arribere & Gilbert, 2013). Moreover, the translation of uORFs may vary in different conditions, including in cancer, thereby modulating the translation efficiency of the corresponding canonical ORF (Young & Wek, 2016; Sendoel *et al*, 2017). Finally, mutations impairing uORFs lead to various human diseases (Barbosa *et al*, 2013; Somers *et al*, 2013).

Beyond 5′ UTR, hundreds of small translated ORFs have also been identified in 3′ UTRs—called downstream (d)ORFs—by ribosome profiling and proteomics (Bazzini *et al*, 2014; Ji *et al*, 2015; Mackowiak *et al*, 2015; Chen *et al*, 2020). However, dORFs have not been systematically characterized and their functions are completely unknown. In this study, we report that dORFs enhance translation of their canonical ORFs in both human cells and zebrafish embryos. Our functional characterization of this effect indicates a novel and strong post-transcriptional regulatory mechanism in vertebrates.

---

1 Stowers Institute for Medical Research, Kansas City, MO, USA
2 Department of Molecular and Integrative Physiology, University of Kansas Medical Center, Kansas City, KS, USA
*Corresponding author. Tel: +1 816 926 4119; E-mail: arb@stowers.org
†These authors contributed equally to this work

# Results

## Translated dORFs are prevalent in vertebrates

To identify potential translated dORFs in vertebrate mRNAs, we first searched the transcriptome of human and zebrafish for in-frame start–stop codon pairs in 3′ UTRs within length of 10–100 codons (Fig EV1A). The most distal ATG was first considered as a possible start codon and then non-ATG (CTG, GTG, TTG) start codons in non-overlapping regions (Fig EV1A). Majority of protein-coding mRNAs contained at least one potential dORF: 82.4% of human and 86.5% of zebrafish mRNAs (Fig 1A). We analyzed ribosome profiling data from human cells (HeLa cells in S phase) (Park *et al*, 2016) and zebrafish embryos at 12 h post-fertilization (hpf) (Bazzini *et al*, 2014) to see whether any of these dORFs are translated and applied an "ORFscore" to each potential dORF. The ORFscore reports active translation based on the 3-nucleotide distribution of the ribosome footprint across a given ORF (Bazzini *et al*, 2014). As shown previously in zebrafish embryos (Bazzini *et al*, 2014), in HeLa cells in S phase (Park *et al*, 2016), annotated canonical ORFs have higher ORFscore than overlapping ORFs (oORF), or ORFs in 5′ UTRs (uORF) and 3′ UTRs (dORF) (Fig EV1B). Of all the possible dORFs in 3′ UTRs, 1,406 and 1,153 displayed evidence of translation in human cells and zebrafish embryos, respectively (Fig 1A, Datasets EV1 and EV2). Since detection of small ORFs is blurred due to their small size and low level of translation, we grouped the translated dORFs into three confidence groups (high, middle, or low) based on their ORFscores (Fig 1A). The translated dORFs were also grouped by their potential translation start codon: ATG and non-ATG, with the non-ATG dORFs comprising more than half of all translated dORFs (Fig EV1C). As a group, dORFs had similar ribosome

footprint distribution compared to canonical ORFs: Ribosomes were enriched at start codons (e.g., human) and stop codons (e.g., zebrafish), distributed relatively uniformly in-frame across the dORF coding region, and depleted upstream and downstream of defined dORF coding regions (Figs 1B and EV1D–F). Overall, dORFs had lower translation levels than previously annotated canonical ORFs (Figs 1B and EV1D). The median size of the translated dORFs was 20 and 19 amino acids in human cells and in zebrafish embryos, respectively (Figs 1C and EV2A), which was significantly shorter than random dORFs that showed no evidence of translation ($P = 3.54e-15$ for human, $P = 7.99e-20$ for zebrafish, Wilcoxon rank-sum test). The sequence between the stop codon of the canonical ORF and start codon of the dORF (Fig 1B), which we refer to as internal UTR (iUTR), had a median length of 105 nucleotides in human and 245 nucleotides in zebrafish (Figs 1C and EV2A). And the uniform distribution of the RNA input reads across the mRNA and in particular between the stop codon of the canonical ORF and the start codon of dORF suggests that dORF is translated from the same mRNA isoform with the canonical ORF (Figs 1B, and EV1D and EV2B).

## The dORF-encoded peptides are often not conserved

Similar to uORFs (Chew *et al*, 2016; Johnstone *et al*, 2016; Couso & Patraquim, 2017), the amino acid sequence of dORFs was not conserved across species (Fig 2A). Of the 1,406 translated dORFs in human, only 6 (0.43%) were conserved and 141 (10.03%) were weakly conserved (Fig 2A). However, similar to uORFs (Johnstone *et al*, 2016), the presence of translated dORFs is conserved in orthologous genes from human and zebrafish (Fig 2B). Specifically, 862 (human) and 610 (zebrafish) genes with translated dORF contain

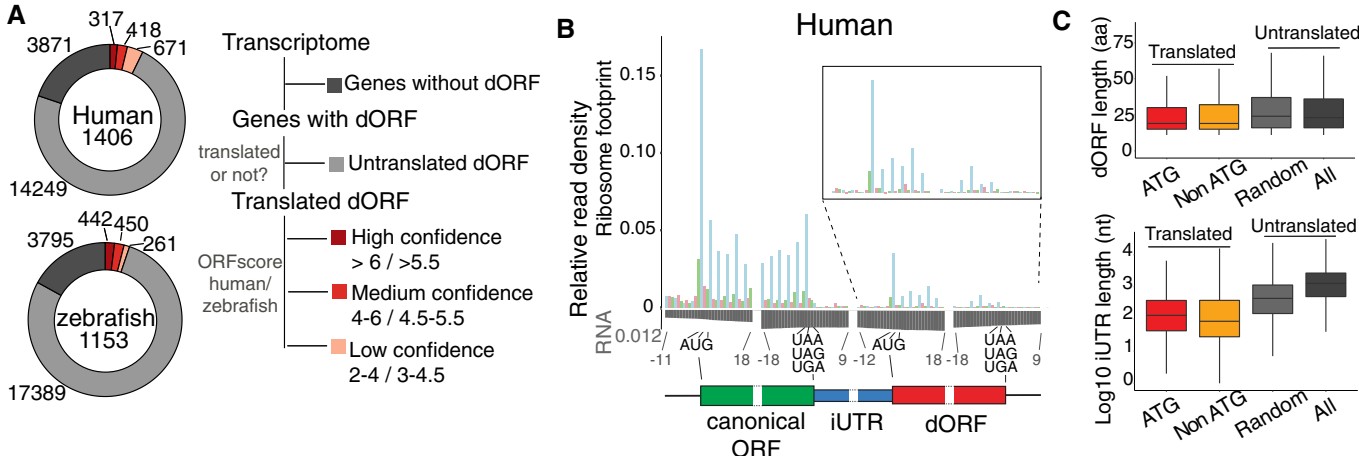

**Figure 1. Translated dORFs are prevalent in vertebrates.**

A Donut plots showing the proportion of transcripts containing high-, medium-, and low-confidence dORFs based on ribosome profiling data from human and zebrafish.

B Metagene plots showing the distribution of the ribosome footprint and input RNA reads around the start and stop codons of canonical ORF and dORF in human high-confidence dORF-containing genes. The ribosome footprint reads mainly show the characteristic 3-nucleotide periodicity across the translated ORF, while the RNA reads are uniform across the transcript. Green indicates the canonical ORF; blue indicates the internal UTR (iUTR), region between the stop codon of the canonical ORF and the start codon of the dORF; red indicates the dORF. Insite shows ribosome distribution at the dORF region, close to its start and stop codons.

C Boxplots showing the lengths of dORFs and iUTR for ATG start dORFs (ATG, red) and non-ATG start dORFs (non-ATG, yellow) with translation evidence, as well as the lengths of random (light gray) and all (dark gray) dORFs for which there was no evidence of translation. The box defines the first and third quartiles, with the median indicated with a thick black line, and vertical lines indicate the variability outside the upper and lower quartiles.

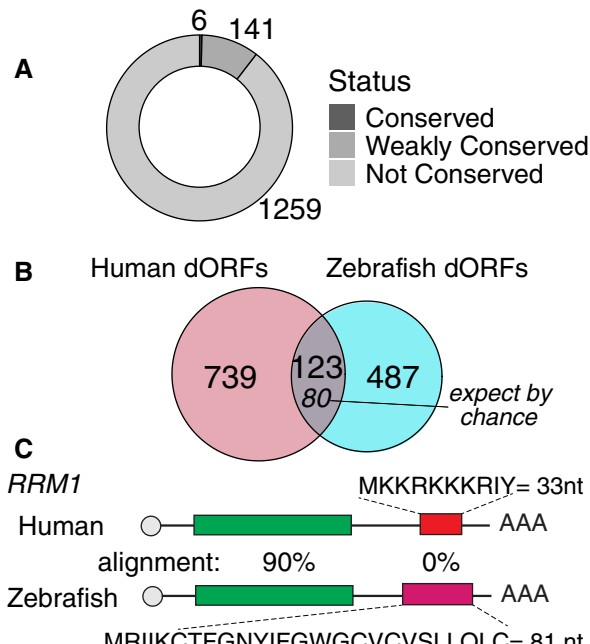

**Figure 2. The dORF-encoded peptides are often not conserved.**

A Donut plot showing the distribution of human dORFs encoding conserved, weakly conserved, and not conserved peptides. Conservation was calculated based on 7-way multiple alignments; dORFs were considered conserved if they had a score > 50, and weakly conserved if they had a score > 0.

B Venn diagram representing the orthologous genes of human and zebrafish in which translated dORFs were identified. The number of dORFs in orthologous genes expected by chance is indicated in italic.

C Cartoon showing *RRM1* gene in human and zebrafish (orthologous), both contain translated dORF, but the dORF sequences are different, as indicated by different colors.

ortholog in another species (Dataset EV3). From those, 123 ortholog genes contained translated dORFs in both species (Fig 2B), showing a significant enrichment compared to a random intersection of approximately 80 genes ($P <$ 2.2e-16, one-sample *t*-test) (Fig EV2C). For example, the orthologous *RRM1* contains translated dORF in human and zebrafish, and while RRM1 canonical ORF is highly conserved (> 90% amino acid alignment), the dORF amino acid sequences are so different that it cannot be aligned (Fig 2C). Moreover, Gene Ontology term analysis of human genes with translated dORFs found enrichment for transcription factors and DNA-binding proteins, when compared to a set of control genes with similar RNA expression levels but no evidence of translated dORFs (Table EV1). Taken together, the significant conservation of dORF presence between human and zebrafish orthologs, despite lack of amino acid conservation, suggests that dORFs likely function through translation activity itself rather than through the polypeptide product.

## mRNAs containing translated dORF are efficiently translated in human cell lines and zebrafish embryos

Translation of uORFs has been shown to repress translation of their canonical ORFs in vertebrates (Johnstone *et al*, 2016). To explore the possible function of dORF, we interrogated the translation efficiency of mRNAs containing translated dORF from ribosome profiling in

human cells (HeLa cell S phase) (Park *et al*, 2016) and zebrafish embryos (12 hpf) (Bazzini *et al*, 2014). Since differences in mRNA level can artificially affect translation efficiency calculation, we resampled the mRNA controls for each uORF and dORF group by choosing genes with similar mRNA levels but no evidence of small ORF translation in either UTR (Figs 3A and EV2D) ($P$ = 0.927, human dORF vs. dORF control; $P$ = 0.806, human uORF vs. uORF; $P$ = 0.82 zebrafish dORF vs. dORF control; $P$ = 0.332, zebrafish uORF vs. uORF, Wilcoxon rank-sum test). After controlling for mRNA levels, we then interrogated the translation efficiency of each group compared to their respective control. As expected, mRNAs containing uORFs had lower translation efficiency than the control group ($P$ = 4.5e-15, human; $P$ = 6.21e-12, zebrafish, Wilcoxon rank-sum test) (Figs 3A and EV2D). Interestingly, mRNAs containing high-confidence ATG-translated dORFs had higher translation efficiency than the controls ($P$ = 1.94e-14, human; $P$ = 0.011, zebrafish, Wilcoxon rank-sum test) (Figs 3A and EV2D). Similarly, enhanced translation efficiency was observed for all groups of dORF-containing genes including medium- and low-confidence translated dORFs (Fig EV2E and F), as well as non-ATG dORFs (Fig EV2F), when compared to their respective control groups. No differences in the regulatory strength were observed between the mRNA containing ATG, GTG, CTG, or TTG-dORFs (Fig EV3A). No significant differences in mRNA stability were observed between dORF-containing transcripts and the control groups in human cells (HeLa cell) (Wu *et al*, 2019) and zebrafish embryos (Bazzini *et al*, 2016) ($P$ = 0.092, human; $P$ = 0.171, zebrafish, Wilcoxon rank-sum test) (Fig EV3B). Similarly, no differences in poly (A) tail length (Chang *et al*, 2014) were observed between dORF-containing transcripts and the control group (Fig EV3C).

Then, we observed that mRNAs containing translated dORFs tend to have shorter canonical ORF ($P <$ 2.2e-16, human; $P$ = 6e-15, zebrafish, Wilcoxon rank-sum test) and longer 3′ UTR ($P$ = 5.5e-14, human; $P$ = 2.9e-14, zebrafish, Wilcoxon rank-sum test) compared to genes without translated dORF (Figs 3B and EV3D). But no significant differences in the 5′ UTR length were observed between the two groups of mRNAs. To exclude the possibility that the observed differences in translation efficiency could be attributed to differences in 5′ UTR, coding, or 3′ UTR lengths (Floor & Doudna, 2016), we resampled control genes without translated dORF, which were chosen to compensate for each of these features ($P$ = 0.25, 5′ UTR length; $P$ = 0.98, main ORF length; $P$ = 0.64, 3′ UTR length for human; $P$ = 0.37, 5′ UTR length; $P$ = 0.68, main ORF length; $P$ = 0.72, 3′ UTR length for zebrafish, mRNA with translated dORF vs. resampled control) as well as similar RNA level (Figs 3C and EV3E). Interestingly, dORF-containing genes still show higher translation efficiency compared to the resampled controls ($P$ = 2.3e-51, 5′ UTR length; $P$ = 1.12e-22, main ORF length; $P$ = 3.38e-39, 3′ UTR length for human; $P$ = 2.04e-12, 5′ UTR length; $P$ = 9.79e-9, main ORF length; $P$ = 1.14e-16, 3′ UTR length for zebrafish, mRNA with translated dORF vs. resampled control) (Figs 3C and EV3E). Together, these results indicate that, in contrast to uORFs, translated dORFs might increase translation of their canonical ORFs.

To verify that the enhanced translation efficiency of mRNAs containing translated dORFs is a consistent feature observed across datasets, we analyzed 28 ribosome profiling datasets from five published studies in different human cells (HEK293T untreated or treated with different drugs, HeLa under cell division phases, human fibroblast uninfected or infected with HSV or

cytomegalovirus) and zebrafish embryos at different hpf (Fig 3D) (Bazzini *et al*, 2014; Rutkowski *et al*, 2015; Sidrauski *et al*, 2015; Tirosh *et al*, 2015; Park *et al*, 2016). As above, control mRNAs for the uORF- and dORF-containing genes were resampled to compare to mRNAs displaying no significant differences in RNA level (Fig 3D, left panel). As expected, mRNAs containing uORFs had lower translation efficiency than the controls in 27 of the 28 datasets, providing further evidence that uORFs have a repressive effect on translation of their canonical ORFs (Fig 3D, right panel, *y*-axis). Interestingly, mRNAs containing dORFs had higher translation efficiencies in all 28 databases analyzed (Fig 3D, right panel, *x*-axis), indicating that the positive effect on canonical ORF translation is a robust and general characteristic of dORFs. Furthermore, the average change in translation efficiency attributable to dORFs is comparable to that observed for uORFs (albeit in opposite direction; median $\log_2$ fold change across samples: 1 for dORFs and −0.5 for uORFs). These regulatory effects are similar in strength to those attributable to microRNAs (Bazzini *et al*, 2012; Johnstone *et al*, 2016) and mRNA modification m6A (Meyer *et al*, 2015). Therefore, dORFs emerge as potential strong and universal regulatory factor in vertebrates, serving to enhance the translation of their canonical ORFs.

## Translation of the dORF is required for enhanced translation of the canonical ORF

To investigate whether active translation of dORFs is required for dORF-mediated enhancement of canonical ORF translation, we compared mCherry expression, from a series of reporters containing endogenous translated dORFs (and iUTRs) to their counterparts in which dORF translation was prevented by insertion of a premature stop codon following the dORF translation start site (dMUT) (Fig 4A). For all four iUTR-dORF sequences tested, mCherry fluorescence intensity was stronger from reporters with intact dORFs than dMUT reporters in human 293T cells with transient DNA transfection; GFP vector was co-transfected as control to normalize transfection difference (Fig 4A). Interestingly, the dORF of *CYR61* starts with TTG, a non-ATG dORF, indicating that, consistent with the genome-wide analysis (Fig EV2F), translation of ATG as well as non-ATG dORFs enhances translation of the canonical ORF.

To confirm that translation itself, rather than the resulted peptide, underlies the observed regulatory effects, we generated mCherry reporters in which endogenous dORFs were replaced with artificially designed dORFs possessing 43 codons (dORF1 and dORF2), in which dORF2 differs from dORF1 by only a single nucleotide insertion which leads to an early frameshift and a completely different codon compositions. As described before, paired control reporters with premature stop codons were also generated for dORF1 and dORF2 (i.e., dMUT1 and dMUT2) (Fig 4B). Both artificial dORF reporters show higher mCherry fluorescence intensity than their translation-deficient counterparts (dMUT1 and dMUT2) with DNA transfection in human 293T cells (Fig 4B). Importantly, a similar effect was seen when *in vitro* transcribed mRNA molecules were transfected directly into the cells, suggesting that this effect is independent of transcription or mRNA processing (Fig 4C). Moreover, minimal differences in mRNA levels were observed between DNA transfection reporter pairs (Fig EV4A), supporting our observation above that dORFs do not affect mRNA stability (Fig EV3B). Importantly, two control iUTRs (*PRMT5* and *TROAP*) from untranslated dORFs based on ribosome profiling did not display significant mCherry fluorescence intensity differences between the dORF1 and their counterparts (dMUT1) (Fig 4B). Therefore, we conclude that, regardless of their nucleotide, codon, or peptide sequences, translation of dORFs enhances canonical ORF translation. One exception to this rule is the iUTR from ribosomal protein L41 (*RPL41*), which shows enhanced mCherry fluorescence intensity when coupled to the endogenous dORF, but not with artificial dORF sequences (dORF1) (Fig EV4B). Moreover, differences in the degree of translation enhancement between different dORFs with the same iUTR (Fig 4) suggest that some sequence features within the dORF itself may impact regulatory strength.

Since functional dORFs can have either ATG or non-ATG start codons (Fig EV1C) according to our genome-wide analyses (Fig EV2F) and reporter assays (Fig 4A), we tested whether the start codon could be replaced. Replacing the "ATG" start codon with any non-ATG codon (CTG, GTG or TTG) within reporter constructs containing the *rrm1* iUTR and artificial dORF1 resulted in higher levels of mCherry fluorescence intensity than in their respective controls (each NTG dMUT) for all non-ATG codons, indicating that alternative start codons are indeed functional (Fig 4D). As expected, no enhanced mCherry fluorescence intensity was observed when the start codon was replaced with a codon that does not initiate translation (AAG), suggesting that translation of the dORF is required. In summary, these results indicate that translation of the dORF is required for its enhancing effect on translation of the canonical ORF (Fig 4).

---

**Figure 3.  mRNAs containing translated dORF are efficiently translated in human cell lines and zebrafish embryos.**

A   Cumulative plot of mRNA level and translation efficiency of genes in human. All genes are indicated in black. Controls for mRNA containing uORF (purple) or dORF (red) were resampled to share similar mRNA level (light purple for uORF controls and orange for dORF controls). *P*-value indicated, Wilcoxon rank-sum test. Only high-confidence ATG dORFs were used in this analysis. Cartoon illustrates while uORFs decrease translation efficiency of the canonical ORF, dORFs increase translation efficiency of the canonical ORF. The data are human HeLa cell S-phase ribosome profiling.

B   Boxplot showing the length of 5′ UTR, CDS (canonical ORF), and 3′ UTR for human genes with translated dORF (red), all genes without translated dORF (dark gray), and resample controls without translated dORF for similar length of the indicated mRNA feature as well as RNA level (light gray). *P*-value indicated, Wilcoxon rank-sum test.

C   Cumulative plot for RNA level and translation efficiency of genes containing translated dORFs in human, controls are genes without translated dORF, which are resampled for similar mRNA level, as well as similar length of either 5′ UTR (left), CDS (middle), or 3′ UTR (right) to compare translation efficiency. *P*-value indicated, Wilcoxon rank-sum test.

D   Scatter plot showing the RNA level (left panel) and translation efficiency (right panel) median $\log_2$ fold change for mRNA containing high-confidence dORFs (ATG + non-ATG) or high-confidence uORFs (ATG) compared to their respect resample control mRNA with neither uORF nor dORF for similar RNA level, across multiple studies. Different samples from the same study show the same color. References and sample conditions are indicated.

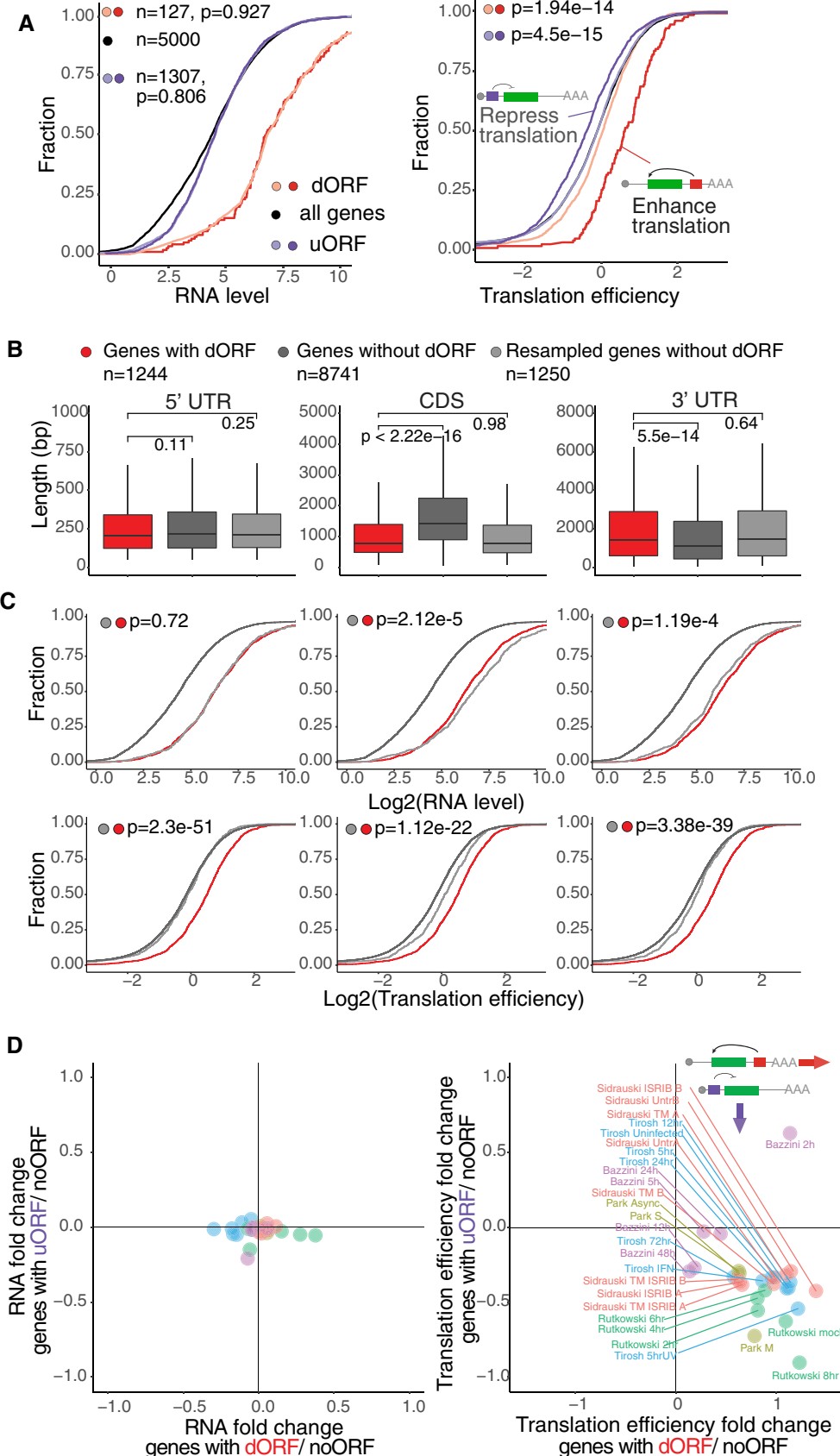

**Figure 3.**

### The number of dORFs, but not dORF length, affects canonical ORF translation

Previous work in human, mice and zebrafish has shown that the number of uORFs in an mRNA correlates with the strength of its translation repression of the canonical ORF (Chew *et al*, 2016; Johnstone *et al*, 2016). Therefore, we interrogated whether an increased number of dORFs also enhance the regulatory effect on canonical ORF translation. In human, 819, 224, and 155 mRNAs contain one, two, or more dORFs, respectively. Compared to resampled control genes with similar RNA levels, all three groups had higher translation efficiency than their control groups (Fig 5A). Moreover, genes with more dORFs had higher translation efficiency than genes with fewer dORFs (3+ vs. 2 dORFs, $P = 0.027$; 2 vs. 1 dORF, $P = 4.15e-05$), indicating that the number of dORFs correlates positively with increased translation efficiency (Fig 5A). And, while there is strong positive correlation between 3′ UTR length and number of all dORFs (translated and untranslated) ($r = 0.413$, $P < 2.2e-16$), there is minimal correlation between 3′ UTR length and number of translated dORF ($r = 0.069$, $P = 0.01$) (Fig EV4C), suggesting that there might

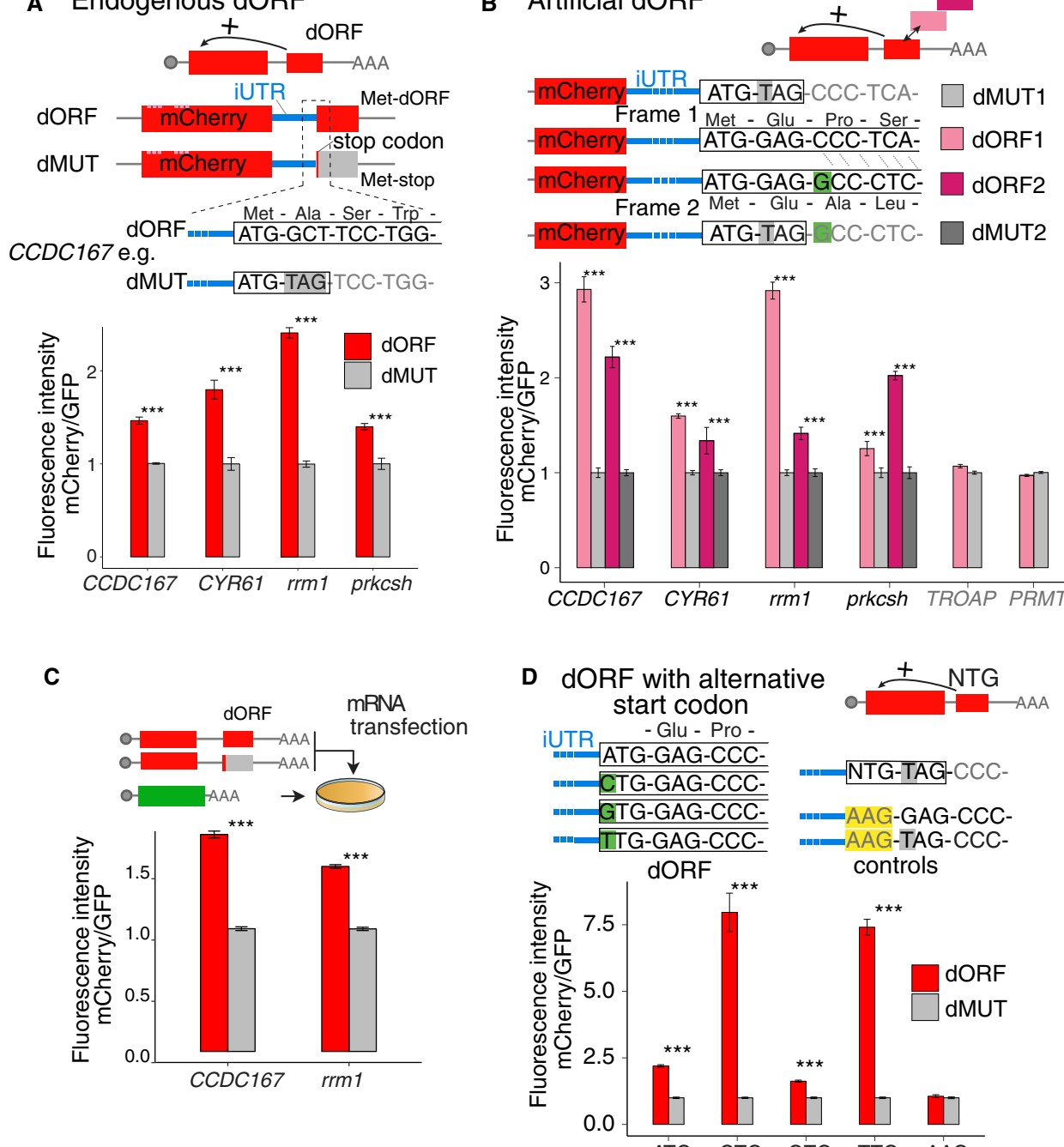

**Figure 4.**

◀

**Figure 4.** **Translation of the dORF is required for enhanced translation of the canonical ORF.**

A  Scheme of paired reporters in which endogenous iUTRs and dORFs from four genes (human *CCDC167* and *CYR61*; zebrafish *rrm1* and *prkcsh*) were cloned downstream mCherry. One to three nucleotide mutations were introduced for each dORF reporter to generate a premature stop codon right after the translation start site (dMUT). Bar plot showing the ratio of fluorescence intensity between mCherry and GFP transfection control in each reporter with DNA transfection, dMUT expression levels were normalized to 1.

B  Diagram of dORF reporters with endogenous iUTRs and artificial dORF sequence. A single G was inserted at beginning of dORF, highlighted in green, to cause frameshift of the dORF reading frame between dORF1 and dORF2. dORF1 and dORF2 have almost same nucleotide composition, but different amino acid sequences. Paired dMUT for each frame was introduced by point mutation to insert a premature stop codon. Bar plot showing fluorescence intensity of dORF and dMUT reporters normalized by GFP transfection control, dMUT expression levels were normalized to 1. All the reporters with iUTRs from translated dORF (*CCDC167*, *CYR61*, *rrm1*, and *prkcsh*) containing artificial dORFs show higher fluorescence intensity than their counterparts (dMUT), regardless of the reading frame or the encoded peptide with DNA transfection. The iUTRs from dORF with no translation evidences (human *TROAP*, *PRMT5*) do not show fluorescence differences between the counterparts (dMUT).

C  *In vitro* transcribed mRNA of dORF and dMUT reporters were transfected into human cells. The iUTRs are from *CCDC167* and *rrm1* with the artificial dORF1 as indicated in Fig 4B. Bar plot showing fluorescence intensity of dORF and dMUT reporters normalized by GFP transfection control, dMUT expression levels were normalized to 1. All the reporters containing artificial dORF show higher fluorescence intensity than their counterparts (dMUT).

D  Illustration of the dORF with alternative start codon. Besides ATG, NTG codons (CTG/GTG/TTG) are also used to replace ATG as dORF start codon (in green); paired dMUT with premature stop codon was also generated for each NTG start codon. As negative control, the codon AAG (in yellow) was used to destroy the start codon of dORF. The *rrm1* iUTR and the artificial dORF1 as indicated in Fig 4B were used. Bar plot shows fluorescence intensity of each reporter normalized by dMUT reporter, all ATG and non-ATG dORFs displayed higher fluorescence intensity than the counterpart controls (dMUT), while no fluorescence intensity difference was observed for the reporter pair with AAG codon.

Data information: For Fig 4, unpaired *t*-test is used ***$P < 0.005$. For cytometry, two biological replicates with two technical replicates were done; the error bar shows SD.

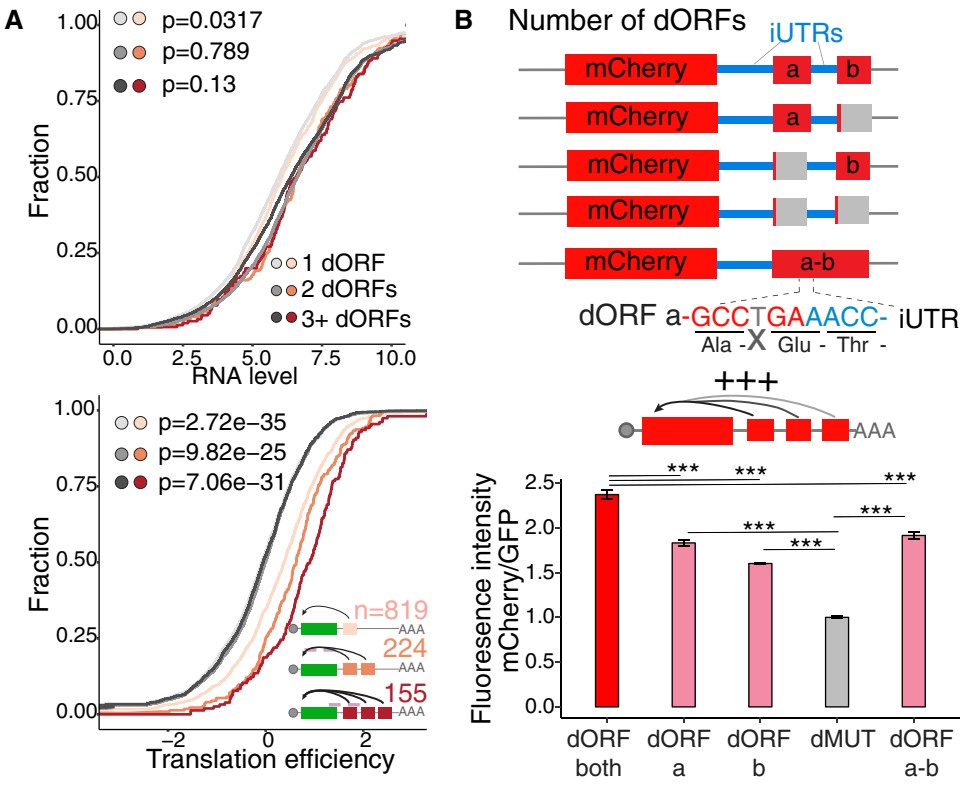

**Figure 5.** **The number of dORFs, but not dORF length, affects canonical ORF translation.**

A  Cumulative distribution of RNA level (top panel) and translation efficiency (bottom panel) of genes containing different numbers of dORFs in human, for each group gene with different numbers of dORFs, controls are resampled for similar mRNA level to compare the translation efficiency. Number of mRNAs and *P*-value indicated, Wilcoxon rank-sum test.

B  Scheme of reporters with different numbers of dORFs. The 3′ UTR of human *E1F1* which originally contains two translated dORFs based on ribosome profiling is cloned downstream of mCherry; premature stop codon in each or both dORF was created by point mutation to change the number of dORF. Additionally, the original stop codon in the first dORF was mutated by deletion of T (indicated in gray) to generate single long dORF. Bar plot showing relative fluorescence intensity of each reporter normalized by GFP transfection control, dMUT expression level is normalized to 1. Reporters with more dORF show stronger enhancing effect for canonical gene expression. Unpaired *t*-test is used ***$P < 0.005$. For cytometry, two biological replicates with two technical replicates were done; the error bar shows SD.

be evolutionary pressure to contain multiple translated dORF in certain mRNAs.

To validate that the increased number of dORFs correlates positively with their enhancing effect on canonical ORF translation, we generated a series of mCherry reporters containing the 3′ UTR of the mRNA for human eukaryotic translation initiation factor 1 (*EIF1*), which contains two translated dORFs according to the ribosome profiling data (23 and 15 amino acids, respectively). mCherry fluorescence intensity was diminished when either of the two dORFs was disrupted by the addition of a premature stop codon (dORFa or dORFb), and further diminished when both dORFs were disrupted (dMUT), suggesting the dORF number has an additive regulatory impact (Fig 5B). To determine whether the number of dORFs or the number of translated codons correlates with enhanced translation, a single nucleotide (T) was removed from the first dORF stop codon creating a single, longer in-frame dORF starting with the first dORF and finishing with the second dORF (58 amino acids) (Fig 5B). Interestingly, this construct containing a single but longer dORF (58 amino acids) resulted in lower mCherry fluorescence intensity when compared to the construct with two dORFs (23 and 15 amino acids, respectively) (Fig 5B). Together, these data indicate that the number of dORFs, but not dORF length, increases the regulatory effect on the translation of the canonical ORF.

### dORFs might be translated by new ribosome recruitment

Since dORFs are located in the 3′ UTR, far from the modified guanosine cap at the 5′ end of the mRNA that recruits ribosome for translation, we hypothesized that dORF translation occurs either by ribosome readthrough following translation termination of the canonical ORF or by new ribosome recruitment from the internal region (Fig 6A). Four lines of evidence suggest that dORFs are not translated by readthrough, but likely by new ribosome recruitment in both human cells and zebrafish embryos. A typical ribosome readthrough has ribosome occupation directly after the stop codon of the canonical ORF as the ribosome continues translation in the same frame (Halvey *et al*, 2012; Dunn *et al*, 2013; Beznoskova *et al*, 2015). None of these two features characterize the translated dORF as a group. First, within the mRNAs containing translated dORFs, the ribosome footprints are depleted in the iUTR region (Figs 1B and EV1D–F). Second, translated dORFs are uniformly distributed in all three reading frames after the stop codon (Fig EV4D). However, it might be possible that one of the ribosome subunits might potentially be involved in a different type of ribosome readthrough (Denis *et al*, 2018). Third, insertion of a 42-nt stem–loop, which prevents ribosome scanning (Jang & Paek, 2016), within the 5′ UTR of a bicistronic reporter, reduces mCherry production from the first ORF, but does not affect GFP production from the second ORF driven by iUTRs from either human *CYR61* or *CCDC167* (Fig 6B and C). Finally, GFP translation driven by the iUTR of *CYR61* was not affected by the insertion of a stem–loop after the stop codon of the mCherry ORF (Fig 6C). These results suggest that iUTRs might work similar to viral internal ribosome entry site (IRES) (Nicholson & White, 2011; Simon & Miller, 2013), where internal ribosome entering site in the iUTR might recruit new ribosome to translate the dORF. To validate that GFP translation was driven by iUTR of *CCDC167* or *CYR61* (Fig 6C) and was not coming from a potential alternative isoform due to promoter or splicing activity of the iUTR sequences, we performed

Northern blot analysis of the bicistronicity reporter (Fig 6D). The absence of shorter isoforms (~ 800 nt) from any of the reporter suggests that these two iUTRs do not have visible transcription capacity. A second band was observed in the SL 5′ reporters in the Northern blot; however, the size of this band (~ 1.4 kb) is larger than that expected by potential iUTR transcription capacity (~ 800 bp), and a single band for the SL 3′ or wt reporter was observed. So, the second band might be due to RNA structure as relative mild denature condition was used. In summary, these data are compatible with the explanation that dORFs are likely translated by new ribosome recruitment and not by ribosome readthrough (Fig 6A).

Therefore, we hypothesized that iUTRs should contain regulatory information recognized by the translation machinery. Indeed, particular positions of the iUTRs as well as the translated dORFs presented significant nucleotide bias compared to the nucleotide composition of the human and zebrafish 3′ UTRs ($P < 0.05$, chi-squared test) (Figs 6E and EV4E). The nucleotide bias tends to be near the translation start codon of the dORF; therefore, we further analyzed the sequence bias at 4-mer level, including three nucleotides upstream and the first nucleotide downstream of the dORF start codon. Bias at the 4-mer level was also observed between translated dORF and dORF with no translation evidences (Figs 6F, and EV5A and B, Datasets EV4 and EV5). The sequence biases were not similar between dORF with different translation start sites (NTG) and/or between human and zebrafish at the nucleotide or 4-mer level (Figs EV4E, and EV5A and B). Moreover, we have not observed strong similarity for the sequence bias nearby start codon between dORF and canonical ORF (Kozak sequence) in human or zebrafish at the nucleotide or 4-mer level (Figs 6G and EV5C). These results suggest that iUTRs contain nucleotide bias that might be required to drive translation.

## Discussion

Our findings support the existence of a previously uncharacterized, strong, conserved, and prevalent post-transcriptional regulatory pathway in vertebrates. Specifically, translation of small ORF in the 3′ UTR, dORF, enhances translation of canonical ORFs within the same mRNA. More than one thousand mRNAs contain translated dORF in human cells and zebrafish embryos. The presence of dORFs in orthologous genes suggests a selective pressure to maintain these dORFs. All groups of mRNAs containing translated dORF displayed higher translation efficiency, independent of their translation start site (ATG, CTG, GTG, or TTC) or of the dORF detection confidence. Specifically, the fact that even the group with the low-confidence dORF detection still displays higher translation efficiency suggests that we might be underestimating the number of mRNAs with translated dORF. The regulatory strength (approximately twofold) of dORF on translation efficiency is comparable to uORFs, microRNAs (Chew *et al*, 2016; Johnstone *et al*, 2016), and m6A-mediated regulation (Meyer *et al*, 2015). We have observed that as a group, mRNAs with translated dORF are consistently efficiently translated across vertebrates under different conditions, however, and similar to uORF (Young & Wek, 2016), it is possible that some dORFs might be regulated in a cell-type- or condition-dependent manner.

The molecular mechanism by which translation of the dORF directly enhances translation of the canonical ORF is unclear. Clues

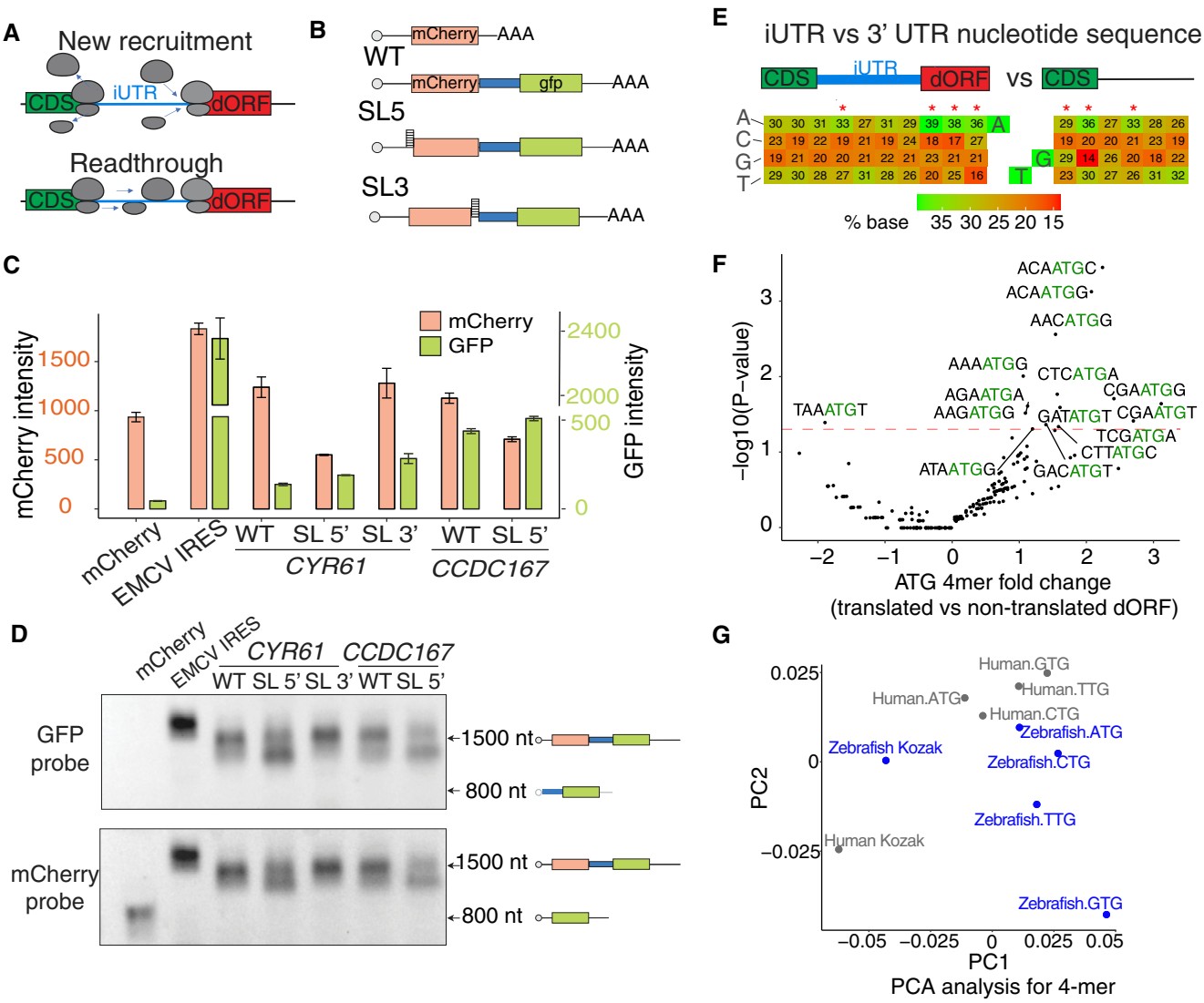

**Figure 6. dORFs might be translated by new ribosome recruitment.**

A Scheme illustrating the dORF translation hypothesis: dORFs may be translated by new ribosome recruitment or by ribosome readthrough after canonical ORF stop codon.

B Scheme of bi-cistronic reporter with a iUTR in the middle. The first ORF mCherry is driven by the cap, while the second ORF GFP might be driven by the iUTR. A 42-nt stem–loop is inserted at the 5′ UTR before mCherry (SL 5′) or between mCherry and iUTR in the 3′ UTR (SL 3′) to inhibit translation.

C Bar plots showing fluorescence intensity of mCherry and GFP in bi-cistronic reporter with *CYR61* and *CCDC167* iUTR. Insertion of stem–loop in 5′ UTR (SL 5′) decreases mCherry fluorescence, while GFP is not affected. Insertion of stem–loop after the stop codon of mCherry (SL 3′) does not decrease the expression of mCherry or GFP. For cytometry, two biological replicates with two technical replicates were done; the error bar shows SD.

D Northern blots of the bi-cistronic reporters showing no alternative splicing or transcription isoforms. Biotinylated DNA oligos anti-GFP and mCherry were used as probes.

E Sequence nearby the dORF start codon (ATG) presented a significant bias compared to the nucleotide composition present in human 3′ UTR. The number shows ratio of each nucleotide in different positions. The red asterisks indicated with position with significant nucleotide bias ($P < 0.05$, chi-squared test).

F Volcano plot showing enrichment/depletion of the 4-mer (three nucleotides upstream and the first downstream of the dORF start codon) between translated and untranslated dORFs for human ATG dORFs (log$_2$, fold change, *y*-axis), and *P*-value (*y*-axis), binomial test.

G PCA for human and zebrafish dORF with different start codons based on the different 4-mer enrichment. Similar analysis was done for the canonical ORF (referred to as Kozak in the figure).

might come from studies of certain viruses that have cap-independent translational enhancers in their 3′ UTRs, which attract translation initiation factors and/or ribosomes to their mRNA to enhance translation (Nicholson & White, 2011; Simon & Miller, 2013). Similarly, tethering experiments have shown that artificial recruitment of the eukaryotic translation initiation factor eIF4G to the 3′ UTR

increases translation of the canonical ORF (Paek *et al*, 2015). Moreover, modification of 3′ UTR regions with N(6)-methyladenosine by the methyltransferase METTL3 can favor mRNA circularization by physically interacting with eIF3h, enhancing translation of the canonical ORF, and promoting oncogenesis (Choe *et al*, 2018). Following the looping model of mRNA structure, where 5′ and 3′

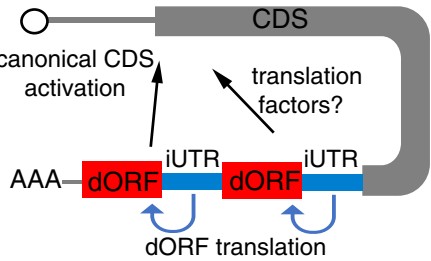

**Figure 7. Model for dORF mechanism.**
iUTR might recruit translation factors and/or ribosomes for dORF translation. Based on the closed loop of mRNA due to 5′–3′ interaction (UTRs crosstalk), the iUTR and dORF at 3′ UTR might be physically closed to the canonical ORF start. Thus, the factors/ribosomes recruited by iUTR for dORF might enhance translation of the canonical ORF, and therefore, higher number of translated dORFs would enhance the regulation strength.

UTRs "crosstalk", recruitment of the translation factors by the iUTR and dORF may also promote translation of the canonical ORF (Fig 7), in a manner similar to that observed by m6A (Choe *et al*, 2018). The observation that the number of dORFs, rather than the length of the dORF, correlates with the regulatory effect suggests that recruitment of translation factors related to initiation might be mechanistically important. The preferred presence of dORF in orthologs with no amino acid conservation and our reporter experiments, in which changing the dORF nucleotides, codons, and encoded amino acids sequence did not abolish the regulatory effect, suggest that it is translation *per se*, and not the dORF-encoded peptide, that is sufficient to enhance translation of the canonical ORF. However, dORFs might have other functions. For example, it was recently shown that two peptides encoded by ORF in 3′ UTR are associated with cell proliferation (Chen *et al*, 2020). Moreover, translation of particular dORFs might be related to mRNA stability or localization.

Finally, our results suggest that dORFs might be translated by ribosome recruitment rather than by readthrough after translation termination of the canonical ORF. And while we have observed significant nucleotide bias closed to the start codon of the translated dORF, it is likely to think that more regulatory information (structural and nucleotide) within the iUTR is encrypted to recruit ribosomes and/or drive translation. Previous work has revealed enriched IRES activity from 3′ UTR regions (Weingarten-Gabbay *et al*, 2016). And different categories of IRES were described in virus based on the translation factors used (Yamamoto *et al*, 2017; Yang & Wang, 2019). The iUTR is relatively short comparing to viral IRES; however, they might share some molecular factors or high-order structure to recruit ribosome. In the future, discovering the regulatory mechanism of dORF activation and the regulatory information driving dORF translation may potentially lead to improved diagnosis of human diseases due to 3′ UTR variants that affect dORF translation (Supek *et al*, 2014), as well as providing potential targets for therapeutic interventions.

# Materials and Methods

## Sample data collection

Published ribosome profiling and RNA-Seq data from zebrafish embryos and human cells were downloaded; zebrafish: Bazzini *et al*

(2014) (SRA314809 and GSE53693); human: Park *et al* (2016) (GSE79664). List of translated dORFs from the other three studies (Rutkowski *et al*, 2015; Sidrauski *et al*, 2015; Tirosh *et al*, 2015) was downloaded from sORFs.org, while processed mRNA and ribosome profiling RPKMs were downloaded from each study (GSE59717, GSE65778, and GSE69906, respectively). For HeLa cell RNA decay data, it was downloaded from Wu *et al* (2019); for zebrafish mRNA half-life, it is downloaded from Bazzini *et al* (2016); for poly(A) tail length of the TAIL-Seq data, it is downloaded from Chang *et al* (2014).

## Sequencing data processing

Reads were mapped to DanRer10, Ens91 (zebrafish), and hg38 (human) genomes. RNA-Seq data were mapped using STAR (–outSAMprimaryFlag OneBestScore; –outFilterMultimapNma× 20; –outFilterMismatchNoverLmax 0.1; –outFilterType BySJout; –alignSJoverhangMin 8; –alignSJDBoverhangMin; –outFilterMismatchNmax 99; –alignIntronMin 20; –alignIntronMax 1000000; –alignMatesGapMax 1000000). Ribosome profiling reads were first trimmed (Fastx_clipper -Q33 -n -z -v) using TGGAATTCTCGGGTGC CAAGG (human) and AGATCGGAAGAGCACACGTCT (zebrafish) adapter sequence. Ribosomal RNAs were eliminated using Bowtie2 and mapped using STAR (–alignEndsType EndToEnd –sjdbScore 2). Only 28–30 and 28–29 bp ribosome footprints were used for human and zebrafish, respectively.

## Translated dORF identification

To identify dORFs with evidence of translation, from DanRer10 and Ens91 (zebrafish) and hg38 and Ens91 (human), the isoform with the most distal stop codon was selected. Potential open reading frames were first defined from the most distal ATG to paired stop codon in all three frames across the entire mRNA. For non-ATG (CTG, GTG, or TTG) dORF, only the annotated 3′ UTR regions were used and non-ATG dORF with an ATG in frame was excluded. Only dORF from 10 to 100 codons were selected. The ORFscore has been run as Bazzini *et al* (2014). For each transcript, dORFs were ordered by decreasing ORFscore and any dORFs overlapping the best dORF were removed. Confidence of translation was defined based on ORFscore. A ORFscore of > 6 in human or > 5.5 in zebrafish was defined as high confidence; > 4/> 4.5 was defined as medium confidence; and > 2/> 3 was defined as low confidence.

## Translation efficiency

Aligned reads were filtered to exclude any genes which had a $\log_2$(RNA level) less than 1 in all time points (human and zebrafish), and translation efficiency was calculated as $\log_2$((rpkm of RPF + 0.05)/(rpkm of RNA level + 0.05)) of the canonical ORF, excluding the first and last coding codons to rule out ORF size effects due to strong peak at the beginning and ends of the ORF. Then, control groups with similar level of RNA to the interrogated groups (e.g., mRNA containing dORF) were generated. Control samples falling below the minimum RNA level of dORF-containing genes were first removed. Then, dORF RNA level was divided into 10 quantiles. Probability of sampling control transcripts from a

given quantile was calculated as the proportion of dORF transcripts within that range divided by the number of control transcripts within that range.

## Conservation of amino acid

dORF transcript coordinates were converted into genomic coordinates in R using the ensembldb package. Spliced dORFs were removed. Coordinates were adjusted so they included only the coding sequence of the dORF (no stop codon). Human 7-way multiple sequence alignment was downloaded from UCSC and uploaded to Galaxy, and multiple alignments were extracted using the Stitch MAF Blocks function. In addition to *Homo sapiens,* the 7-way alignment included *Pan troglodyes, Macaca mulatta, Canis lupus familiaris, Mus musculus, Rattus norvegicus,* and *Monodelphis domestica.* The output from Galaxy was split such that one file contained multiple alignments for one dORF; then, all files were scored using PhyloCSF with the following parameters: hg38.7way –strategy omega –files [filelist] –minCodons = 10 –removeRefGaps. dORFs were considered conserved if they had a score of > 50, and weakly conserved if they had a score > 0.

## Conservation of dORF presence

Human orthologs of zebrafish transcripts were downloaded from Ensembl version 91. First, we defined a set of 9,242 one-to-one human–zebrafish orthologs where each ortholog was present in our dORF analysis. Then, the number of genes which contained translated dORF in both species was determined. To determine whether this intersection was significant, genes were randomly sampled from human and zebrafish equal to the number of transcripts containing dORFs in each species. This was repeated 1,000 times, with the intersection of each random group being determined at each iteration, the 95% confidence interval was determined, and the resampled group was compared to the true intersection using a *t*-test.

## GO analysis

The GO analysis is done on Gorilla website (Eden *et al*, 2009), with running mode of two unranked lists of genes. The human dORF-containing genes are used as target; human resampled control genes with similar mRNA level but no translated dORF are used as background.

## Metagene plot

The only read length used for the metaplots was as follows: 28–30 nt for human and 28–29 nt for zebrafish. Only the top 2,000 highly expressed transcripts in terms of reads per kilobase per million (RPKM) were used. These transcripts must also satisfy CDS, 5′ UTR, and 3′ UTR length all greater than or equal to 50 bp. To estimate RPKM expression, reads were trimmed to the 5′-most base. Next, we shifted the 5′- and 3′-position of each transcript by −12 and −15 bp, respectively. The two windows (each with length 100 bp) were chosen for the metagene plots around the coding start and end positions. For each transcript, read counts were computed for each position in the two windows and then scaled by the total number of

reads within the two windows. The metagene plots were produced by taking the mean normalized read counts of each position in the two windows for all transcripts selected. In this way, the plots were not biased toward transcripts with extremely high read counts. Also, the two windows are interdependent after normalization to the same scale, making them directly comparable.

## Initiation context

For human and zebrafish genes with translated dORFs, we separate them by different start codons. The 20 nucleotides surrounding each dORF were extracted, and frequencies of nucleotides at each position were determined. To determine whether a nucleotide has significant bias at a given position, the nucleotide frequencies were compared to the frequency of nucleotides in the 3′ UTR as control of each species using a chi-squared test. Enrichment analysis of the 4-mer consisting of the three nucleotides upstream and one nucleotide downstream of the dORF start codon was performed by first determining frequencies of 4-mers in each start codon in human and zebrafish. 4-mer frequencies were compared to frequencies of the same 4-mer in untranslated dORFs of the same start codon as control using a binomial test. PCA was done for both the 4-mer frequency and nucleotide bias nearby dORF start codon. Kozak sequence (sequence nearby canonical ORF start codon) frequencies were included as a reference for translation initiation in either species.

## Tissue culture

293T were cultured with DMEM media, supplied with 10% FBS, L-glutamine, and penicillin/streptomycin. The cells were ordered from tissue culture facility from the Stowers Institute, at relatively low passage, lower than passage 15.

## Cloning and DNA transfection

All the cloning to insert iUTR or dORF after mCherry was done by Gibson assembly with NEBuilder HiFi DNA Assembly Master Mix following protocols. This will avoid any gap between mCherry-iUTR or iUTR-dORF. Sequence information is included in the expanded view tables and also snapgene files. For DNA transfection in 293T cells, it was transfected with Lipofectamine 3000 based on manufacturer's instruction. The plate is set overnight before transfection in 24-well plate, so the cells are around 70% confluent the day for transfection. 500 ng total DNA per well was added with transfection reagents. 24 h post-transfection, cells are collected for cytometry or RNA extract. For cytometry, we have two biological replicates × 2 technique replicates. For RNA analysis, we have two biological replicates × 3 technique replicates.

## *In vitro* transcription and RNA transfection

The plasmid constructs containing dORF-related reporters were linearized with Not1HF; similarly, GFP-containing control plasmid was also linearized with Not1HF. To generate normal cap and poly(A) mRNA, linearized plasmids were then *in vitro* transcribed using SP6 mMessage mMachine Kit (Life technology), following *in vitro* polyadenylated with Poly(A) Tailing Kit from Ambion. mRNA was purified by Qiagen RNeasy Mini Kit, and the RNA concentration was quantified by Qubit

RNA Broad Range Kit. For RNA transfection, cells were transfected with TransIT®-mRNA Transfection Kit from Mirus Company based on manufacturer's instruction in 24-well plate with 500 ng total RNA per well. 24 h post-transfection, cells are collected for cytometry.

### Cytometry analysis

The florescent reporter intensity of the cells was quantified in ZE5 equipment, using laser of GFP (488/510) and mCherry (587/610), cells were suspended in DMEM with 10% FBS for running cytometry. Cells were not fixed. Cytometry data.fsc file was analyzed with FlowJo, median intensity of the cells was used to represent fluorescent intensity.

### Northern blot

Total RNA from 24-well plate was extracted with TRIzol chloroform, and the RNA was suspended with 20 μl nuclease-free H2O. The concentration of RNA was measured by Qubit, to load around 500 ng RNA per well. RNA gel running was following the protocol from Lonza Bioscience. Total RNA was resuspended with 1× MOPS buffer, formaldehyde, and deionized formamide. Heat at 70°C for 10 min, chill on ice, and add loading buffer before running. Then, RNA was migrated using 1× MOPS at 100 V for 3 h and transferred with 10× SSC overnight. Oligonucleotide DNA probes with 3′-biotin were ordered from IDT with HPLC purification. Probing and detection were done following the protocol of North2South® Chemiluminescent Hybridization and Detection Kit from Thermo Fisher, using streptavidin–HRP. The probe for mCherry is cctttctgatgacgcttcccat cccattgt-/3Bio, and the probe for GFP (zsgreen) is CTTGGAC TCGTGGTACATGCA GTTCTCCTC-/3Bio/.

## Data availability

The sequence, genomic position, and other information of the translated dORF in human and zebrafish are available in Datasets EV1 and EV2. The list of orthologous genes (Human and Zebrafish) that contain translated dORF in both species is available in Dataset EV3. The frequency and enrichment of the 4-mer (three nucleotides upstream and the first one downstream of the translation start site of the translated dORF in human and zebrafish) are available in Datasets EV4 and EV5. The sequence of the iUTR and/or dORF used in the reporters is available in Source Data for Expanded View.

**Expanded View** for this article is available online.

## Acknowledgements

We thank Dr Kausik Si, Dr Carter Takacs, Dr Yi Liu, and Life Science Editors for suggestions and critical reading of the manuscript. We also thank Joan and Ron Conaway for their generous support. We thank the following Stowers core facilities: Bioinformatics, Molecular Biology, Cytometry, and Tissue Culture. We also thank all Bazzini laboratory members for their help. This study was supported by the Stowers Institute for Medical Research. A.A.B was awarded with a Pew Innovation Fund. This work was performed as part of thesis research for QW, Graduate School of the Stowers Institute for Medical Research. Original data underlying this manuscript can be accessed from the Stowers Original Data Repository at: http://www.stowers.org/research/publications/libpb-1497.

## Author contributions

QW, MW, and AAB designed and conceived the project and interpreted the data. QW performed most of experiments and also analyzed the data. MW analyzed most of the ribosome profiling data. WDB provided support for experiments. MMG and NZ helped with ribosome profiling data analysis. QW and AAB wrote the manuscript with input from the other authors.

## Conflict of interest

The authors declare that they have no conflict of interest.

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
