## [Review Process File · The EMBO Journal]

Translation of small downstream ORFs enhances translation of canonical main open reading frames

Ariel Bazzini, Qiushuang Wu, Matthew Wright, Madelaine Gogol, William Bradford, and Ning Zhang

DOI: [10.15252/emboj.2020104763](https://doi.org/10.15252/emboj.2020104763)

Corresponding author(s): Ariel Bazzini (arb@stowers.org)

Review Timeline:

Submission Date:	19th Feb 20
Editorial Decision:	25th Mar 20
Revision Received:	19th May 20
Editorial Decision:	16th Jun 20
Revision Received:	23rd Jun 20
Accepted:	26th Jun 20

Editor: Stefanie Boehm

Transaction Report:

Thank you for submitting your manuscript proposing an effect of downstream ORFs on translation of the corresponding canonical ORF for consideration by The EMBO Journal. We have now received three reports on your study, which are included below for your information.

As you will see, the referees express interest in the proposed model, but also raise several issues that would need to be addressed in a revised version. In particular, they find further detail regarding the assay system would need to be provided (ref#1 point 2, ref#3 general points 1, 2, 3, specific points 3, 4), as well as clarifying which dataset was used for the analysis in Fig. 2B (ref#3 point 2). Moreover, the use of the proposed confidence groups should be reassessed/ further discussed (ref#2 general point 4), as well as the role of mRNA isoforms (ref#1 point 1). In addition to these key points, I would ask you to also carefully respond to all other comments and revise the manuscript accordingly.

Addressing the referees' comments fully would likely require a significant amount of experimental work and time and we realize that an extensive revision may at this point in time not be feasible due to the COVID-19/SARS-CoV-2 pandemic affecting labs worldwide. While we can extend the reviewing deadlines and have extended our 'scooping protection policy' to cover the period required for a full revision, we strongly encourage you to contact us to discuss a detailed revision plan to ensure that potential issues, including those due to the current circumstances, are addressed before committing to a revision.

Thank you again for giving us the opportunity to consider your manuscript, I look forward to hearing from you.

REFEREE REPORTS

Referee #1:

Wu and colleagues examine the presence and functions of ORFs in the 3' UTRs of human and zebrafish protein-coding mRNAs. Using a number of high-throughput data sets, the authors nicely demonstrate the widespread nature of these downstream ORFs (dORFs) and suggest that dORF translation enhances translation of their corresponding canonical ORFs. Reporter plasmids nicely support and confirm this observation, including the interesting finding that the number of dORFs - and not the length or peptide they encode-is key for the translation stimulatory effect. The manuscript is overall very clear, well written, and will be of broad interest.

(1) The authors have provided a few Northern blots to help rule out the alternative hypothesis that differences in the mRNA isoforms are responsible for the phenotypes observed, but they should ideally show RNA analysis for all the reporter experiments (Figs 3 and 4). In Fig 1G and S3C, there is a smaller band on the Northern blots with the SL 5' constructs that the authors do not address. A single isoform is seen with the SL 3' construct in 1G so I think the authors' interpretations are correct, but it would be really nice to further rule out the presence of alternative isoforms in the other analyses presented. I will point out that the mRNA transfection experiment in the supplemental material somewhat (but not fully) helps address this concern.

(2) To promote reproducibility, please provide sequences of reporter plasmids used throughout the manuscript, especially the exact sequences of iUTRs and WT/mutated dORFs used in Figs 3 and 4.

Minor points:

(1) When describing Fig. S6B in the main text, it is more appropriate to write that there were "minimal" differences in mRNA levels rather than "no differences."

(2) The authors should re-read the manuscript for grammatical accuracy. There are some incorrect tenses used at times. Also, the authors should be careful with how mCherry is written - they often write it with a capitalized C in the figures, but not always.

(3) Figure S6D should be mentioned in the main text, e.g. in 2nd paragraph of discussion section.

(4) In Fig 1, the authors may want to adjust the red/green coloring scheme to avoid confusion. In 1B/E, green in the main ORF and red is the dORF, but the reverse is true in 1F.

Referee #2:

This manuscript uncovers a very interesting, and so far unappreciated, gene regulation mechanism: downstream ORFs (dORFs). My general impression is that this work is important and has great potential to open a new field. However, it suffers from several, yet surmountable, shortcomings. The text is very assertive, in my opinion beyond what the results warrant. I suggest a number of experiments and controls that would allow the authors to maintain their assertions. Alternatively, they may just want to add some essential controls, and tone down their language from "dORFs enhance translation of canonical/main ORFs" to "dORFs can enhance translation of..." throughout the text.

A general comment is that, due to the format as a short communication, the manuscript becomes very tedious to read, having to flick from main to Supplementary figures without great justification other than space-saving. I would present whole experiments/set of results in either main or supplementary, rather than half the data in each. The figure legends are too telegraphic and one

needs to flick between main text, methods, main figure and supp. Fig. legends to understand how an observation was made, and what was the whole set of results obtained.

Most of the analysis focuses on the most distal dORFs. This should be made clear (may be referring to "distal dORFs" or "ddORFs" when appropriate).

Another general comment is that mRNAs are translated as circles (as mentioned in the discussion), that recycle ribosomes (hence Rbs) from the 3'UTR to the cap. Thus, any alteration of the 3'UTR can have an effect on the translation rate of 5' ORFs. The authors should carefully think of this when interpreting their results.

Finally the authors should consider what is gained by dividing their translated dORFs into high, medium and low confidence sets. The graphs look very similar, and often only the high confidence set is used in calculations and comparisons. Currently the proliferation of graphs is confusing and looks like an attempt to inflate the data, when in fact it would be much better to include human and zebrafish data, side by side in the main figures. If the authors are so convinced that translation of dORFs increase main ORF translation, they should define a cut-off, justify it, and stick with it. Perhaps they could "reverse-engineer" and use only the high-confidence set, because it is the one that shows clearer correlation with main ORF translation. In this way, the paper would claim that "high translation of dORFs affects canonical ORF translation" a concept that many more people would be happy to accept (as opposed to the current "every and all translated dORFs have strong effects on main ORF translation").

-page 3:

Please merge figures 1A and S1B.

-what is the size of non-translated dORFs?

-The mechanism for dORF translation the authors are referring to is reinitiation and main ORF readthrough, I believe (this page and following). Neither is cap-independent. The alternative ribosome recruitment mechanism they propose is already known as IRES, and should be referred as that.

-What is the size of the stem loop introduced? Loops do not "preclude" but delay/impede ribosome scanning (see authors own data) : mRNAs make loops anyway, which the Ribosome unzips. It can have an effect on re-initiation just because its length (if the distance between main ORF and dORF goes above 2-300bp).

Of the 4 lines of evidence presented favouring IRES activity, only the third and fourth (the dicistronic experiments) would be relevant, and yet some extra controls would be desirable (see below). A) Footprints would also be reduced in the iUTR under re-initiation (as scanning Rbs would not get stuck at poor codons). B) Rbs scan in the three frames so these footprints would also be expected to give a such a signal under re-initiation.

Thus the text should indicate that the presented results are compatible with IRES activity of the iUTRs, and IRES-mediated translation of dORFs, not that they prove such activity. Even if all optional controls provided below were included, there would still be 2 translated dORFs assessed out of 1,406 detected.

Alternative/Optional: the SL3 construct should also be done for the CCDC167 gene. For both genes, a loop should be introduced just 5' to the dORF. There is no indication otherwise that the

loop could affect the dORF (i.e. we have no positive control). Also, another control is that these experiments carried out with a gene containing a non-translated dORF. Finally, the size of the loops (or rather, extra sequences inserted) should increase the distance between main ORF and the next downstream ORF, and between the dORF and its next 5' ORF, to above 300bp, to preclude re-initiation more conclusively.

-Page 4:

Optional: I intuitively agree that the overlap of conserved translated dORFs is > random, yet I do not think that the logic and the method used is the best one. The authors may wish to: 1) define single-orthologue pairs Homo-Danio, 2) establish frequency of translated and untranslated ddORFs in each species in this set, and overlaps (both species in a pair translated). 3) carry out Chi square or t-test (according to frequencies). Currently, due to extra paralogues in Danio, ddORFs have extra-chances to score as 'conserved', and we anyway can't compare canonical genes conservation to ddORF conservation, as shown by the authors own results showing low conservation of ddORFs.

Essential: 1) please justify PhyloCSF cut-offs. Amend last sentence, second paragraph to "most dORFs likely function...". As conclusively shown recently by Chen et al. Science 2020, a putative prevalent function for a ORF class as cis-regulators (uORFs) does not preclude a minority functioning as peptide-producing.

2) Amend first sentence, third para. to "Presence of uORF can reduce translation..." (in line with page 7) or else, expand the citations to include other works that had shown a repressive effect of uORFs before Johnstone 2016.

-Page 5:

Throughout this page, correlations are treated as causal relationships supporting the authors' hypothesis that dORFs enhance main ORF translation, when in fact it could be the reverse (highly translated main ORFs 'leak' Rbs and hence translation onto their 3'UTR, see optional control below) or yet a third unknown factor driving both indicators.

Essential: text should be amended to say that the correlations are compatible with these explanations, not that they "indicate/prove/do/show" any of them.

The observed moderate TE changes (up to 2-fold) can easily arise through minor ribosomal occupancy and RNA level differences when both values are low to begin with, since the division $TE = RPKM/RNA$, will amplify the values. The authors indicate towards the end of the first paragraph that "the control set also had low TEs" and their TE calculation includes the unusual $RPKM+0.05$ formulae, perhaps to include $RPKM=0$ values in their calculations. $RPKM=0$ is not acceptable for RNA values, and these should be removed from the calculations. Some cut-off to distinguish actual transcription and translation from background noises should be used.

-line 6-7: "Observed differences in translation efficiency could not be attributed to differences in 5'UTR, coding or 3'UTR lengths (Fig. S5C)". Fig S5C does not include information on translation efficiency. Perhaps the authors mean Fig. S5D? Looking at plots and p-values, Fig. S5C seems to denote significant differences between 3'UTR lengths containing translated and untranslated dORF groups in both cases. Figure S5D then shows that the cumulative translation efficiency between dORFs contained in 3'UTRs of similar length is higher for the translated group. This indicates that 3'UTR length doesn't directly influence translational efficiency. However, longer 3'UTRs could have more sequence space to accommodate translated dORFs, which could

indirectly lead to an overall increase in translational efficiency of the canonical ORF. In Figure S5C, the 3'UTRs of mRNAs with translated dORFs are larger than those of controls for both human and zebrafish canonical genes. Does the number of dORFs correlate with the length of 3'UTRs? Perhaps a panel correlation between 3'UTR length and dORF number (absolute and translated) would be helpful.

Finally, a statistical difference between populations does not predict the behaviours of each member of the population. The last sentence of the second paragraph is too sweeping and should be amended.

Optional: To discard that the increased TE of dORFs is due to increased TE of canonical ORFs, we authors might want to generate a novel construct similar to those in 1F but with lowered (or abolished) main ORF translational efficiency (though mutation of mCherry Kozak or AUG), and observe relative changes in TE of GFP-dORF

Page 6:

-I could not find out how many transfections were carried out, nor the statistical test used to give significance to changes in normalised intensities from 1 to <1.5 (Fig. 3A,B). It is also disturbing that the differences between dORF1 and dORF2 are in some cases much higher than between them and their respective dMUT. Optional: the authors could consider an alternative assay, such as quantitative Westerns, to validate at least some crucial experiments.

-Second para. I guess the substituted ddORF was also 43 nt long?

Page 7:

-last sentence, second para., amend to "indicating that the number of dORFs correlates positively with increased translation efficiency".

-First sentence of the discussion is too assertive. It should rather read "Our findings support the existence of a previously uncharacterised post-transcriptional regulation mechanism in vertebrates" or similar. As the discussion indicates, we have no idea how this works, and whether it includes a new molecular pathway.

Typos etc. (quite numerous-please revise the manuscript):

Main Text:

- page three, second sentence should read "3-nucleotide codon distribution" or similar to clarify the biological significance of the ORF score; and second to last line: "could coming" - change to "could be coming".
- page four, second para. line 4 should read "...the presence of translated dORFs is conserved.." And under section "mRNAs containing translated...", line 5: "can artificially affects" - remove "s".
- page five, under "Translation of the dORF is required...", line 1 - "To investigate whether activate" - change to "To investigate whether active"

Figures:

- Figure 1B, the colours of the gene diagram should be different that those used for read density.
- Position of Panels 1G and 1H/I should be switched. Clarify what species were included in the 7-way PhyloCSF study.
- Figure 2A legend, last sentence should read "cartoon illustrates that while uORFs associate with

lower translation efficiency, dORFs associate with higher one" or similar, without extracting mechanistic information not warranted by the data.

- Figure 4 legend: Re-format last sentence as it contains a smaller font size.

- Figure S5: red corresponds to genes containing translated dORFs while gray corresponds to genes with no dORFs or rather genes containing untranslated dORFs? These are alternatively called "control genes" in - Legend S5A, "control genes without dORF" in Legend S5C and "genes without translated dORF" in Legend Are these different sets of controls? Please clarify.

Materials and Methods:

- Page 22: "Conservation of presence" should read "Conservation of dORF presence"

I finish by repeating that this manuscript, properly revised, should provide a landmark in the field of gene regulation, and that my comments should not be construed as reasons to reject the paper. I congratulate the authors and recommend that they tone down their language or carry out the extra experiments suggested; there is no "right to forget" in scientific publishing, and the main message (dORFs can act as translational regulators) would remain, and be more solid.

Referee #3:

EMBO J-2020-104763

Wu et al., manuscript

"Translation of small open reading frames in 3' UTRs enhances translation of their canonical open reading frames"

There are two parts to this paper one is analysis based, which appears reasonable, but this is not my area of expertise. However, in summary, the authors have analysed ribosome profiling data from a human (Park et al., 2016) and a zebrafish embryo dataset (Bazzini et al., 2014). They found that about 82% of human mRNAs and about 86% of zebrafish mRNAs have a potential 10 - 100aa ORFs in the 3'UTR (dORFs) and of all dORFs 1406 and 1153 showed evidence of translation in human dataset and zebrafish embryo dataset, respectively. The translated dORFs had both AUG start codon (47%) and non-AUG start codons (CUG, GUG, UUG) and the median size of the dORFs was about 60nt. The region between STOP of the canonical ORF and start of the dORF is called internal UTR (iUTR) and in human dataset the median iUTR length was about 105nt and about 245nt in zebrafish embryo dataset. The ribosome footprint distribution was similar in the dORFs compared to the canonical ORFs, although dORFs had lower translation levels than previously annotated canonical ORFs. However, their analysis showed that unlike translation of uORFs in the 5'UTR, which decreased translation of the canonical ORFs, translation of dORFs seemed to enhance translation of their canonical ORFs. To confirm these data they analysed an additional 28 ribosome profiling datasets from five published studies in human and zebrafish.

In the experimental part of the project to show that indeed the translation of the dORFs were needed for the increased translation efficiency of the canonical ORF, they fused iUTR-dORF from four mRNAs to mCherry vector and compared these to paired dMUT constructs, in which they had mutated the dORF by inserting a premature STOP codon after the start codon so that the dORF was not translated. These experiments were by DNA transfections in HEK 293T cells. They also showed with mCherry constructs that the translation of dORF was important rather than the peptide sequence of the dORF. However a significant amount of additional work needs to be carried out to really conclude that this very unusual finding is correct, and the comments below need to be addressed.

General points:

1. It is essential to show an alignment of the translated iUTR-dORF sequences. This would allow the reader to assess if there were anything common between the iUTR-dORFs that are translated. The alignments could be divided into the three categories; high, medium and low confidence. These alignments would also show the context of the potential start codon of the translated dORFs, which would be important to assess.
2. Nucleotide sequence of the iUTR-dORFs that they use in their mCherry experiments have to be included in the main manuscript.
3. The M&M section does not contain any information of how the cloning was carried out for either the mCherry-GFP vector or to the mCherry vector, this information should be added.

Specific points:

1. Fig 1; The authors show evidence that the dORFs are translated by recruiting new ribosomes after the STOP codon of the canonical ORF. They stated that the median length of the iUTR in human dataset was ~105nt, which seems very short for this to function as an internal initiation site. To show that the iUTR sequence is really important for the translation of dORFs, the authors need should generate constructs where the reverse iUTR sequence is used in their bi-cistronic mCherry-GFP vector.
2. Fig 2; The datasets used in the analysis in Fig 2B are obtained from experiments in which human cells have been either treated with tunicamycin (causing ER stress, Sidrauski) or infected with cytomegalovirus/HSV1 (Rutkowski/Tirosh). I assume that this means that the ribosome profiling was carried out from both non-treated and treated/infected cells and that the authors used all the datasets in their analysis. The authors need to analyse the nontreated and treated/virus infected datasets separately to determine if stress has an effect on the dORF translation and enhancement of their canonical ORF translation.
3. Fig 3; (a) Paired constructs in which endogenous iUTR-dORFs from two human mRNAs were cloned downstream to mCherry vector (CCDC167 and CYR61) and from two zebrafish embryo mRNAs (rm1 and prkcsh) were used in the DNA transfection experiments in 3A. Which confidence group do these dORFs belong to? Do all these dORFs have the same start codon? (b) The human CCDC167 iUTR-dORF construct gives much higher mCherry fluorescent in the 3B than in the 3A, why would this be? The other constructs show similar results in both experiments. (c) Which iUTR-dORF sequence was used in the 3C? The start codon of the dORF was mutated from AUG to non-AUG codon and CUG/UUG codons seemed to work much better than AUG. However, the authors do not comment on the big difference between the wt (=AUG) and CUG which is about 7.5 x fold change to the CUG-dMUT (or the UUG/UUG-dMUT). Is this because the CUG-dMUT/UUG-dMUT fluorescence is less than AUG-dMUT or is the CUG/UUG-dORF really so much better translated?
4. Fig 4; Their analysis also showed that number of dORFs had a positive effect on the canonical ORF translation. In the panel 4B the iUTR-dORFs from eIF1 mRNA was cloned to the mCherry vector. The sequence of the iUTR and the dORFs needs to be shown.
5. Suppl FigS6; The panel S6D seems to have a putative model of how the authors think the dORF might enhance translation of canonical ORFs, but they do not refer to this figure in the text nor do they discuss how the dORF translation might enhance translation of the canonical ORF.

Referee #1:

Wu and colleagues examine the presence and functions of ORFs in the 3' UTRs of human and zebrafish protein-coding mRNAs. Using a number of high-throughput data sets, the authors nicely demonstrate the widespread nature of these downstream ORFs (dORFs) and suggest that dORF translation enhances translation of their corresponding canonical ORFs. Reporter plasmids nicely support and confirm this observation, including the interesting finding that the number of dORFs - and not the length or peptide they encode-is key for the translation stimulatory effect. The manuscript is overall very clear, well written, and will be of broad interest.

(1) The authors have provided a few Northern blots to help rule out the alternative hypothesis that differences in the mRNA isoforms are responsible for the phenotypes observed, but they should ideally show RNA analysis for all the reporter experiments (Figs 3 and 4). In Fig 1G and S3C, there is a smaller band on the Northern blots with the SL 5' constructs that the authors do not address. A single isoform is seen with the SL 3' construct in 1G so I think the authors' interpretations are correct, but it would be really nice to further rule out the presence of alternative isoforms in the other analyses presented. I will point out that the mRNA transfection experiment in the supplemental material somewhat (but not fully) helps address this concern.

We thank the reviewer for this suggestion and while we agree that it would have been ideal to perform northern-blot for each reporter experiment in original Fig 3 and Fig 4 (now Fig 4 and Fig 5). It is important to mention that although different mRNA isoforms might potentially be responsible for detection of some translated small ORF, the hypothesis of an alternative isoforms are NOT likely to explain the phenomenon that translation of dORF enhances translation of the canonical ORF (Original Fig 3 and Fig 4). Specifically, if we hypothesize that the reporters (Original Fig 3 and Fig 4) could potentially express two isoforms, the large one driven by the promoter and a potential smaller one driven by the iUTR, assuming that the iUTR could potentially have transcription activity. However, if that would be the case, the single mutations (premature stop codons, Original Fig 3 and Fig 4), or the mutations in the translation start sites (NTG, original Fig 3C, now Fig 4D) should not affect the translation of the canonical ORF, they should only affect the translation of the small ORF in the potential shorter isoform. And while it is possible that a single mutation in the 3'UTR can affect translation of the canonical ORF, we also show that that a single mutation replacing the translation start site (NTG) for AAG, indeed abolishes the enhanced translation between the "wild-type" and mutant counterpart with a premature stop codon.

Moreover, as the reviewer indicated our experiment with RNA transfection supports that increased translation of the canonical ORF is driven by translation of the dORF within the same mRNA, not by transcription or mRNA processing. Following the same rationale, the alternative isoforms hypothesis is unlikely to explain the effect in translation efficiency that we observed across the 28 ribosome profiling.

However, the alternative isoform hypothesis is important to consider for detection of translated dORF in 3'UTR. Therefore, we have re-analyzed the RNA-seq data from ribosome profiling input to determine potential isoforms from the endogenous mRNA containing translated dORF. Specifically, we have:

-Plotted the RNA input in a metaplot in parallel with ribosome footprint in Fig 1B and Fig EV1D. Now, in Fig 1B, we show a uniform distribution across the mRNA and specifically between the stop codon of the canonical ORF and the iUTR-dORF region. It suggests that there are not alternative isoforms coming from the iUTR-dORF regions.

-While the metaplot is a well-established way of showing the distribution of the reads across the mRNA, we also did a more sensitive analysis comparing the fold change of the read coverage

from the region close the stop codon of the canonical ORF and regions upstream and downstream from each mRNA. We hypothesize that if alternative isoforms are present in the iUTR-dORF regions, then the fold change of RNA coverage comparing a region upstream the stop codon of the canonical ORF and iUTR-dORF region should be different. In Fig EV2B, we show similar fold change coverage between in stop codon of the canonical ORF and different regions of the mRNA with and without translated dORF. This result also suggests that the iUTR-dORF are likely in the same isoform with canonical ORF.

With respect to the second band in the northern-blot seen with SL 5' (now Fig 6D), we have used relative mild denature condition. So, we think that the second band could be due to the mRNA structure, especially when the stem loop is at the very end of mRNA. Moreover, the size of the mentioned band is larger (around 1.4kb) than the one predicted to be driven by the iUTR (around 800nt). In the SL 3' and wt reporter, we see just one band, so it is not likely that the 2nd band in SL 5' is due to promoter activity. However, we really want to thank the reviewer comments, and we have discussed it in the text.

(2) To promote reproducibility, please provide sequences of reporter plasmids used throughout the manuscript, especially the exact sequences of iUTRs and WT/mutated dORFs used in Figs 3 and 4.

This is a very crucial point for us, now we provide all the sequences in Expanded View table.

Minor points:

(1) When describing Fig. S6B in the main text, it is more appropriate to write that there were "minimal" differences in mRNA levels rather than "no differences."

We agree, we have modified it.

(2) The authors should re-read the manuscript for grammatical accuracy. There are some incorrect tenses used at times. Also, the authors should be careful with how mCherry is written - they often write it with a capitalized C in the figures, but not always.

Thanks, we have checked the entire manuscript.

(3) Figure S6D should be mentioned in the main text, e.g. in 2nd paragraph of discussion section. We agree with the reviewer's comments, we have brought this to the main Fig 7, and we mention it in the discussion.

(4) In Fig 1, the authors may want to adjust the red/green coloring scheme to avoid confusion. In 1B/E, green in the main ORF and red is the dORF, but the reverse is true in 1F.

We thank this suggestion; we changed the color in Fig 6 (part of old Fig 1)

Referee #2:

This manuscript uncovers a very interesting, and so far unappreciated, gene regulation mechanism: downstream ORFs (dORFs). My general impression is that this work is important and has great potential to open a new field. However, it suffers from several, yet surmountable, shortcomings. The text is very assertive, in my opinion beyond what the results warrant. I suggest a number of experiments and controls that would allow the authors to maintain their assertions. Alternatively,

they may just want to add some essential controls, and tone down their language from "dORFs enhance translation of canonical/main ORFs" to "dORFs can enhance translation of..." throughout the text.

We want to thank the reviewer for her/his opinions about our work, we agree that this work "has great potential to open a new field". And we also agree to tune down our language.

A general comment is that, due to the format as a short communication, the manuscript becomes very tedious to read, having to flick from main to Supplementary figures without great justification other than space-saving. I would present whole experiments/set of results in either main or supplementary, rather than half the data in each. The figure legends are too telegraphic and one needs to flick between main text, methods, main figure and supp. Fig. legends to understand how an observation was made, and what was the whole set of results obtained.

Her/his comments let us think about this point and we totally agree with this suggestion. We have re-organized the figures. And we also re-wrote the figure legends.

Most of the analysis focuses on the most distal dORFs. This should be made clear (may be referring to "distal dORFs" or "ddORFs" when appropriate).

We thank the reviewer for mentioning this point. We want to clarify it. When we defined all potential small ORFs in the 3'UTR (translated or none-translated), we first consider the most distal ATG from each stop codon based on only the nucleotide sequence (please see Fig EV1A. However, when we define translated dORF based on ribosome profiling, it does not need to be the 1st potential dORF or the most distal dORF. The translated dORF could be any of the small ORF across the 3'UTR, as long as they have ribosome footprint to meet the requirement. So, we have never considered the distal dORF when considering the dORF activation. However, we modified the text to clarify this point. Although we cannot assume that all dORFs starts with the codons defined, in the metaplot of Fig 1B and Fig EV1D-F for ribosome footprint, we get reads enrichment in the defined start codon (ATG/NTG), indicating that are likely the ones used for most dORFs.

Another general comment is that mRNAs are translated as circles (as mentioned in the discussion), that recycle ribosomes (hence Rbs) from the 3'UTR to the cap. Thus, any alteration of the 3'UTR can have an effect on the translation rate of 5' ORFs. The authors should carefully think of this when interpreting their results.

We totally agree with her/his comment. And that is the main reason why we have created a control for each single reporter, thus we could minimize the mutations/insertions to precisely control the experiments. For example, for the reporters used in Fig 4A, the dORF and dMUT counterparts only differed in 1, 2 or 3 nucleotides. In Fig 4 B-D, the only difference between each of the 11 dORF reporters compare to their counterparts is a single mutation. Moreover, in Fig 4B we introduced a single insertion changing the frame but not the coding capability and observed the dORF effect. In Fig 4D, we modified the translation start codon (NTG start codon) and we observed the dORF regulation comparing to each of the paired respective controls (1 nucleotide mutation). However, when we introduced a single mutation affecting the translation start site, (AAG is not a translation start codon) we did not observe the dORF effect indicating that translation of the dORF is important. We have modified the text and properly discussed it.

Finally the authors should consider what is gained by dividing their translated dORFs into high, medium and low confidence sets. The graphs look very similar, and often only the high confidence set is used in calculations and comparisons. Currently the proliferation of graphs is confusing and

looks like an attempt to inflate the data, when in fact it would be much better to include human and zebrafish data, side by side in the main figures. If the authors are so convinced that translation of dORFs increase main ORF translation, they should define a cut-off, justify it, and stick with it. Perhaps they could "reverse-engineer" and use only the high-confidence set, because it is the one that shows clearer correlation with main ORF translation. In this way, the paper would claim that "high translation of dORFs affects canonical ORF translation" a concept that many more people would be happy to accept (as opposed to the current "every and all translated dORFs have strong effects on main ORF translation").

We thank the reviewer for bringing this point. Small ORF detection is very hard, since they are small and usually poorly translated. So, we like the idea of separating them in confident groups. We have previously taken similar approach studying upstream ORFs (upORF or uORF) in zebrafish (Johnstone et al EMBO Journal, 2016, PMID: 26896445). Moreover, as the reviewer indicated all the groups behave similarly, in all the groups we have observed that the canonical ORF presented higher translation efficiency than the control. For us, it is important to show that the regulatory dORF effect is across the different confident groups because if we just group all of them together as a single big group, then, the regulatory effects might be coming only from the most highly confident ones. Therefore, the fact that we still observed increased translation efficiency in the least confident groups, indicates that we are not over-calling the number of translated dORF. However, now we included in the discussion that not necessary "every and all translated dORFs have strong effects on canonical ORF translation". This is a very important point because we cannot rule out that dORF might have other activity or role (protein function, mRNA quality control, etc).

Please merge figures 1A and S1B.

We have modified these panels accordingly.

-what is the size of non-translated dORFs?

We apologize if it was not clear enough, in the original Fig 1C we plotted the length distribution of a random set of non-translated dORFs, the median size for untranslated dORF is 69nt. We re-plot the Fig 1C including length of "ALL the non-translated", however, the distribution is very similar to the random set.

-The mechanism for dORF translation the authors are referring to is reinitiation and main ORF readthrough, I believe (this page and following). Neither is cap-independent. The alternative ribosome recruitment mechanism they propose is already known as IRES, and should be referred as that.

We appreciate the suggestion and included in the text.

-What is the size of the stem loop introduced? Loops do not "preclude" but delay/impede ribosome scanning (see authors own data) : mRNAs make loops anyway, which the Ribosome unzips. It can have an effect on re-initiation just because its length (if the distance between main ORF and dORF goes above 2-300bp).

The stem loop we use is from previous publication, with 42 nt in total.

Of the 4 lines of evidence presented favouring IRES activity, only the third and fourth (the dicistronic experiments) would be relevant, and yet some extra controls would be desirable (see below). A) Footprints would also be reduced in the iUTR under re-initiation (as scanning Rbs would not get stuck at poor codons). B) Rbs scan in the three frames so these footprints would also be expected to give a such a signal under re-initiation.

Thus the text should indicate that the presented results are compatible with IRES activity of the iUTRs, and IRES-mediated translation of dORFs, not that they prove such activity. Even if all optional controls provided below were included, there would still be 2 translated dORFs assessed out of 1,406 detected.

We agree that the 3rd and 4th evidences from the reporter assays are the most direct evidence indicating that dORFs are translated by new ribosome recruitment, similar to IRES and not by ribosome readthrough. However, based on (Dunn, 2013; Halvey, 2012 and Beznoskova, 2015), typical readthrough has ribosome occupation directly after stop codon and decrease with distance, the ribosomes keep the same frame after the stop codon of the main ORF and continue translation. It is true that the iUTR also works like IRES for internal ribosome recruitment, while we have originally mentioned it, now we increase our discussion about it.

It is also true that we have tested two iUTRs in bi-cistronic reporter for its translation activity and compare to the large number of dORF detected we cannot generalize the observation. This is clearly one question that rises from this work and we plan to tackle it in the future: what is the regulatory information in the iUTR driving translation of the dORF. As the reviewer suggests we need to analyze several hundreds of them to be able to generalize. Therefore, we modified the text accordingly and reduce our statement strength.

Alternative/Optional: the SL3 construct should also be done for the CCDC167 gene. For both genes, a loop should be introduced just 5' to the dORF. There is no indication otherwise that the loop could affect the dORF (i.e. we have no positive control). Also, another control is that these experiments carried out with a gene containing a non-translated dORF. Finally, the size of the loops (or rather, extra sequences inserted) should increase the distance between main ORF and the next downstream ORF, and between the dORF and its next 5' ORF, to above 300bp, to preclude re-initiation more conclusively.

Thanks for these suggestions. In the future, we plan to dissect the regulatory information of the iUTR and these are all great experiments. We really appreciate her/his comments.

-Page 4:

Optional: I intuitively agree that the overlap of conserved translated dORFs is > random, yet I do not think that the logic and the method used is the best one. The authors may wish to: 1) define single-orthologue pairs Homo-Danio, 2) establish frequency of translated and untranslated ddORFs in each species in this set, and overlaps (both species in a pair translated). 3) carry out Chi square or t-test (according to frequencies). Currently, due to extra paralogues in Danio, ddORFs have extra-chances to score as 'conserved', and we anyway can't compare canonical genes conservation to ddORF conservation, as shown by the authors own results showing low conservation of ddORFs.

Essential: 1) please justify PhyloCSF cut-offs. Amend last sentence, second paragraph to "most dORFs likely function...". As conclusively shown recently by Chen et al. Science 2020, a putative prevalent function for a ORF class as cis-regulators (uORFs) does not preclude a minority functioning as peptide-producing.

2) Amend first sentence, third para. to "Presence of uORF can reduce translation..." (in line with page 7) or else, expand the citations to include other works that had shown a repressive effect of uORFs before Johnstone 2016.

Thanks for all these suggestions. While we originally followed the same approach that we used to study upstream ORF (uORF) conservation (Johnstone 2016 EMBO Journal). Now, we re-analyzed our data following the feedback from the reviewer. We observed similar results, the presence of translated dORF is conserved between human and zebrafish. We have decided to include the reviewer's analysis rather than the original to rule out any potential effect of dealing with multiple orthologues for a given gene.

We also want to thank the reviewer mentioning the very recent paper from the Weissman lab (Chen et al. Science 2020), which was published after our initial submission. We have incorporated it into the discussion.

Finally, the reviewer is absolutely right, there are several papers showing the uORF can reduce translation across multiple species. We have expanded the citations.

-Page 5:

Throughout this page, correlations are treated as causal relationships supporting the authors' hypothesis that dORFs enhance main ORF translation, when in fact it could be the reverse (highly translated main ORFs 'leak' Rbs and hence translation onto their 3'UTR, see optional control below) or yet a third unknown factor driving both indicators.

Essential: text should be amended to say that the correlations are compatible with these explanations, not that they "indicate/prove/do/show" any of them.

We agree and modified the text accordingly.

The observed moderate TE changes (up to 2-fold) can easily arise through minor ribosomal occupancy and RNA level differences when both values are low to begin with, since the division $TE = RPKM/RNA$, will amplify the values. The authors indicate towards the end of the first paragraph that "the control set also had low TEs" and their TE calculation includes the unusual $RPKM+0.05$ formulae, perhaps to include $RPKM = 0$ values in their calculations. $RPKM = 0$ is not acceptable for RNA values, and these should be removed from the calculations. Some cut-off to distinguish actual transcription and translation from background noises should be used.

The reviewer is absolutely right about the artefactual effect of analyzing translation by ribosome profiling of poorly expressed mRNA. As we have indicated in the method part: aligned reads were filtered to exclude any genes which had a $\log_2(\text{RNA level})$ less than 1 in all time points (human and zebrafish). Moreover, we were extremely careful with the analysis, as it can be seen in the cumulative plots from Fig 3 and 5 as well as in Fig EV2 and 3. mRNAs containing dORFs are highly express (RNA level). Moreover, as control we resample mRNAs with no detected translated dORF but similar level of mRNA. Therefore, our control groups are also highly expressed. Therefore, we do not have the potential problem that the reviewer mention. However, we modified the text because it is not clear enough. Rather than saying that the control shows low TEs, we should have said the mRNA containing translated dORF displayed higher TE. One more time, we want to thank the reviewer comments because it helps us clarifying the message.

-line 6-7: "Observed differences in translation efficiency could not be attributed to differences in 5'UTR, coding or 3'UTR lengths (Fig. S5C)". Fig S5C does not include information on translation efficiency. Perhaps the authors mean Fig. S5D? Looking at plots and p-values, Fig. S5C seems to denote significant differences between 3'UTR lengths containing translated and untranslated dORF groups in both cases. Figure S5D then shows that the cumulative translation efficiency between dORFs contained in 3'UTRs of similar length is higher for the translated group. This indicates that 3'UTR length doesn't directly influence translational efficiency. However, longer 3'UTRs could have

more sequence space to accommodate translated dORFs, which could indirectly lead to an overall increase in translational efficiency of the canonical ORF. In Figure S5C, the 3'UTRs of mRNAs with translated dORFs are larger than those of controls for both human and zebrafish canonical genes. Does the number of dORFs correlate with the length of 3'UTRs? Perhaps a panel correlation between 3'UTR length and dORF number (absolute and translated) would be helpful.

We apologize for this misunderstanding. Thanks for the reviewer comments, we have decided to include this data as part of main figure because it rules out potential bias in the analysis. Now, for the length of 5'UTR/CDS/ 3'UTR, we show 3 group of genes rather than 2 groups: genes with translated dORF, genes without dORF, and we added the resampled controls for genes without dORF but with similar length of 5'UTR or CDS or 3'UTR and RNA level. Therefore, it is easy to observe that genes with dORF might have different length for each gene region. However, after controlling for length and RNA level (old Fig S5C, now Fig 3C), we show that still mRNA with dORF displayed higher translation efficiency.

A more precise sentence should be: Observed differences in translation efficiency (now Fig 3B) could not be attributed to differences in 5'UTR, coding or 3'UTR lengths (Fig 3C)".

Moreover, we have added new analysis about 3'UTR length and dORF number in Fig EV4C. While there is a strong correlation between 3'UTR length and number of all potential dORFs (translated and untranslated) ($r=0.4$, $p < 2.2e-16$), we observed that there is weak correlation between 3'UTR length and number of translated dORF ($r=0.069$, $p = 0.01$).

Finally, a statistical difference between populations does not predict the behaviours of each member of the population. The last sentence of the second paragraph is too sweeping and should be amended.

We have changed it.

Optional: To discard that the increased TE of dORFs is due to increased TE of canonical ORFs, we authors might want to generate a novel construct similar to those in 1F but with lowered (or abolished) main ORF translational efficiency (though mutation of mCherry Kozak or AUG), and observe relative changes in TE of GFP-dORF

Thanks for the suggestion, it is a good control that we will consider in the future.

Page 6:

-I could not find out how many transfections were carried out, nor the statistical test used to give significance to changes in normalised intensities from 1 to <1.5 (Fig. 3A,B). It is also disturbing that the differences between dORF1 and dORF2 are in some cases much higher than between them and their respective dMUT. Optional: the authors could consider an alternative assay, such as quantitative Westerns, to validate at least some crucial experiments.

For the cytometry analysis, we have used two biological replicates with two technical replicates each. For the qPCR analysis, we have used two biological replicates with three technical replicates each. We modified the method section to make it clear. The statistics are indicated in the legend part. We used unpaired two tail t.test to do the statistics for reporter assays.

We think that the quantitative western-blot can be interesting, however our data suggests that those differences are not simple detection noise because the errors bars are minimal. We think that the regulatory information encoded in the dORF (including the translation start site) might impact regulatory strength. For example, the initiation context of each alternative translation start site looks to be different (New Fig 6), and while all of NTG substitution work, the degree could be different.

-Second para. I guess the substituted ddORF was also 43 nt long?

It is also 43aa. We have clarified the text and legend.

Page 7:

-last sentence, second para., amend to "indicating that the number of dORFs correlates positively with increased translation efficiency".

We have changed it accordingly.

-First sentence of the discussion is too assertive. It should rather read "Our findings support the existence of a previously uncharacterised post-transcriptional regulation mechanism in vertebrates" or similar. As the discussion indicates, we have no idea how this works, and whether it includes a new molecular pathway.

We have changed it accordingly.

Typos etc. (quite numerous-please revise the manuscript):

Main Text:

- page three, second sentence should read "3-nucleotide codon distribution" or similar to clarify the biological significance of the ORF score; and second to last line: "could coming" - change to "could be coming".

- page four, second para. line 4 should read "...the presence of translated dORFs is conserved.." And under section "mRNAs containing translated...", line 5: "can artificially affects" - remove "s".

- page five, under "Translation of the dORF is required...", line 1 - "To investigate whether activate" - change to "To investigate whether active"

Figures:

- Figure 1B, the colours of the gene diagram should be different that those used for read density.

- Position of Panels 1G and 1H/I should be switched. Clarify what species were included in the 7-way PhyloCSF study.

-Figure 2A legend, last sentence should read "cartoon illustrates that while uORFs associate with lower translation efficiency, dORFs associate with higher one" or similar, without extracting mechanistic information not warranted by the data.

- Figure 4 legend: Re-format last sentence as it contains a smaller font size.

- Figure S5: red corresponds to genes containing translated dORFs while gray corresponds to genes with no dORFs or rather genes containing untranslated dORFs? These are alternatively called "control genes" in - Legend S5A, "control genes without dORF" in Legend S5C and "genes without translated dORF" in Legend Are these different sets of controls? Please clarify.

Materials and Methods:

- Page 22: "Conservation of presence" should read "Conservation of dORF presence"

Thanks for pointing these out. We change all of them accordingly.

The 7-way phyloCSF study includes *Homo sapiens*, *Pan troglodyes*, *Macaca mulatta*, *Canis lupus familiaris*, *Mus musculus*, *Rattus norvegicus*, and *Monodelphis domestica*, and we have added to the method.

I finish by repeating that this manuscript, properly revised, should provide a landmark in the field of gene regulation, and that my comments should not be construed as reasons to reject the paper. I congratulate the authors and recommend that they tone down their language or carry out the extra

experiments suggested; there is no "right to forget" in scientific publishing, and the main message (dORFs can act as translational regulators) would remain, and be more solid.

We want to thank the reviewer for such constructive comments and we also really appreciate her/his comments about our work.

Referee #3:

EMBO J-2020-104763

Wu et al., manuscript

"Translation of small open reading frames in 3' UTRs enhances translation of their canonical open reading frames"

There are two parts to this paper one is analysis based, which appears reasonable, but this is not my area of expertise. However, in summary, the authors have analysed ribosome profiling data from a human (Park et al., 2016) and a zebrafish embryo dataset (Bazzini et al., 2014). They found that about 82% of human mRNAs and about 86% of zebrafish mRNAs have a potential 10 - 100aa ORFs in the 3'UTR (dORFs) and of all dORFs 1406 and 1153 showed evidence of translation in human dataset and zebrafish embryo dataset, respectively. The translated dORFs had both AUG start codon (47%) and non-AUG start codons (CUG, GUG, UUG) and the median size of the dORFs was about 60nt. The region between STOP of the canonical ORF and start of the dORF is called internal UTR (iUTR) and in human dataset the median iUTR length was about 105nt and about 245nt in zebrafish embryo dataset. The ribosome footprint distribution was similar in the dORFs compared to the canonical ORFs, although dORFs had lower translation levels than previously annotated canonical ORFs. However, their analysis showed that unlike translation of uORFs in the 5'UTR, which decreased translation of the canonical ORFs, translation of dORFs seemed to enhance translation of their canonical ORFs. To confirm these data they analysed an additional 28 ribosome profiling datasets from five published studies in human and zebrafish.

In the experimental part of the project to show that indeed the translation of the dORFs were needed for the increased translation efficiency of the canonical ORF, they fused iUTR-dORF from four mRNAs to mCherry vector and compared these to paired dMUT constructs, in which they had mutated the dORF by inserting a premature STOP codon after the start codon so that the dORF was not translated. These experiments were by DNA transfections in HEK 293T cells. They also showed with mCherry constructs that the translation of dORF was important rather than the peptide sequence of the dORF. However a significant amount of additional work needs to be carried out to really conclude that this very unusual finding is correct, and the comments below need to be addressed.

General points:

1. It is essential to show an alignment of the translated iUTR-dORF sequences. This would allow the reader to assess if there were anything common between the iUTR-dORFs that are translated. The alignments could be divided into the three categories; high, medium and low confidence. These alignments would also show the context of the potential start codon of the translated dORFs, which would be important to assess.

We want to thank the reviewer for this suggestion. This a very interesting point, we did the analysis and the new results are shown in Fig 6 and Fig EV4-5. We have found that translated dORF contains nucleotide bias, especially close to the dORF translation start site. We have also observed bias at the 4-mer level (3 upstream and 1 downstream of the translation start site). The bias between species and translation start sites does not look to be similar. Moreover, the bias looks to be different to the

one observed for canonical ORFs (Kozak sequence). This analysis will be the starting point to dissect the regulatory information of the iUTR driving translation of the dORF.

2. Nucleotide sequence of the iUTR-dORFs that they use in their mCherry experiments have to be included in the main manuscript.

Now, we provide all the sequence in Expanded View table and maps (SnapGene).

3. The M&M section does not contain any information of how the cloning was carried out for either the mCherry-GFP vector or to the mCherry vector, this information should be added.

Now, we provide all the sequence in Expanded View table; we also add detailed information for cloning in the method part. For short answer, we did Gibson cloning for all the vectors.

Specific points:

1. Fig 1; The authors show evidence that the dORFs are translated by recruiting new ribosomes after the STOP codon of the canonical ORF. They stated that the median length of the iUTR in human dataset was ~105nt, which seems very short for this to function as an internal initiation site. To show that the iUTR sequence is really important for the translation of dORFs, the authors need should generate constructs where the reverse iUTR sequence is used in their bi-cistronic mCherry-GFP vector.

We thank for this suggestion. The reverse sequence is a very common control for DNA regulatory elements, however for RNA the reverse sequence would be unrelated with the sense and therefore it can be taken as a random sequence. Therefore, while the reviewer control is a good one, we have included two iUTRs from human genes, with no experimental translation evidence (ribosome profiling) as negative controls (Part of New Fig 4B).

2. Fig 2; The datasets used in the analysis in Fig 2B are obtained from experiments in which human cells have been either treated with tunicamycin (causing ER stress, Sidrauski) or infected with cytomegalovirus/HSV1 (Rutkowski/Tirosh). I assumes that this means that the ribosome profiling was carried out from both non-treated and treated/infected cells and that the authors used all the datasets in their analysis. The authors need to analyse the nontreated and treated/virus infected datasets separately to determine if stress has an effect on the dORF translation and enhancement of their canonical ORF translation.

We also thanks for this suggestion, this is also an extremely interesting point. As it was shown for uORF, dORF might also be expressed in a cell, tissue or condition specific manner. However, the detection of the small ORF (including dORF) highly depends on the quality of the ribosome profiling. Therefore, several further validations would be required to demonstrate the potential role of dORF under stress condition. Although this is fascinating suggestion indicating the potential biological role of dORF, we think that this is beyond the scope of this paper, but we will definitely explore this in the future. When we analyzed ribosome profiling from different studies, we did separate each condition to see dORF effect: different human cells (HEK293T untreated or treated with drug IRSIB, HeLa under cell division phases, human fibroblast uninfected or infected with HSV or Cytomegalovirus), and zebrafish embryos at different hpf (now Fig 3D). Now, we indicate the sample identity of each of the 28 ribosome profiling and we have enlarged the discussion about it.

3. Fig 3; (a) Paired constructs in which endogenous iUTR-dORFs from two human mRNAs were cloned downstream to mCherry vector (CCDC167 and CYR61) and from two zebrafish embryo mRNAs (rm1 and prkcs) were used in the DNA transfection experiments in 3A. Which confidence group do these dORFs belong to? Do all these dORFs have the same start codon?

The iUTRs we have tested came from high confident groups. CYR61 has TTG as start codon, while others have ATG.

(b) The human CCDC167 iUTR-dORF construct gives much higher mCherry fluorescent in the 3B than in the 3A, why would this be? The other constructs show similar results in both experiments. The result of Fig 3B (now Fig 4B) is with artificial dORF sequence, while Fig 3A (now Fig 4B) contains endogenous dORF of each iUTR. We hypothesize that the sequence differences might explain the variation.

(c) Which iUTR-dORF sequence was used in the 3C? The start codon of the dORF was mutated from AUG to non-AUG codon and CUG/UUG codons seemed to work much better than AUG. However, the authors do not comment on the big difference between the wt (=AUG) and CUG which is about 7.5 x fold change to the CUG-dMUT (or the UUG/UUG-dMUT). Is this because the CUG-dMUT/UUG-dMUT fluorescence is less than AUG-dMUT or is the CUG/UUG-dORF really so much better translated?

We apologize that we did not make it clear in the text. The iUTR is from zebrafish *rrm1*, and the dORF sequence is artificial dORF1. And while all the translation start codon confer the increased translation of the canonical ORF, they might not work equally in the same context. Following the reviewer comments, we have re-analyzed the ribosome profiling, as a group for each NTG dORFs (ATG/CTG/GTG or TTG), the canonical ORF displayed comparable high translation efficiency (New Fig EV3A). We have not observed significant difference with respect to the regulation strength considering the translation start codon.

4. Fig 4; Their analysis also showed that number of dORFs had a positive effect on the canonical ORF translation. In the panel 4B the iUTR-dORFs from eIF1 mRNA was cloned to the mCherry vector. The sequence of the iUTR and the dORFs needs to be shown.

Now, we provide all the sequences in Expanded View table, and also Snap-Gene files.

5. Suppl FigS6; The panel S6D seems to have a putative model of how the authors think the dORF might enhance translation of canonical ORFs, but they do not refer to this figure in the text nor do they discuss how the dORF translation might enhance translation of the canonical ORF.

We appreciated her/his suggestion, now we made it Fig 7, and also discussed the model.

Thank you for submitting your revised manuscript for our consideration. Please apologize the delay in communicating this decision to you, which was due to delayed referee reports and the high number of new submission we are currently receiving. We now have the reports from the original referees (see comments below). I am pleased to say that all referees now support publication. Referee # 1 points out some minor textual issues that can be resolved in a final revised version. In this version, I would also ask you to please address a number of editorial issues that are listed in detail below. Please make any changes to the manuscript text in the attached document only using the "track changes" option. Once these remaining issues are resolved, we will be happy to formally accept the manuscript for publication.

Thank you again for giving us the chance to consider your manuscript for The EMBO Journal. I look forward to receiving your final revision. Please feel free to contact me if you have further questions regarding the revision or any of the specific points listed below.

REFeree REPORTS

Referee #1:

The authors have satisfactorily addressed my concerns.

Minor corrections to the text:

- (1) The authors often refer to "HeLa cells in stage S", which I assume means the cells are in S phase of the cell cycle? I think more readers would understand the term "S phase".
- (2) Top of p.6: Fig. 3D should be 3C.
- (3) P.10, Line 6: EV4D should be EV4E.

Referee #2:

I think that the current manuscript fully addresses my concerns and suggestions, and those of the other referees. I believe that the paper is ready to be accepted, and that we should spare the authors further rounds of revision that could complicate the paper for minimal gain. It already contains a lot of cutting-edge work, carried out to a very high standard.

JUAN PABLO COUSO

Referee #3:

While the authors have not carried out all the experiments that I requested, the majority of the comments made have been adequately addressed. This is a really important study and will be of interest to many in this field.

Corresponding Author Name: Ariel Bazzini

Manuscript Number: 2020-104763